# Theoretical Analysis of Auto Rate-tuning by Batch Normalization

**Sanjeev Arora**
Princeton University and Institute for Advanced Study
arora@cs.princeton.edu

**Zhiyuan Li**
Princeton University
zhiyuanli@cs.princeton.edu

**Kaifeng Lyu** *
Tsinghus University
lkf15@mails.tsinghua.edu.cn

## Abstract

Batch Normalization (BN) has become a cornerstone of deep learning across diverse architectures, appearing to help optimization as well as generalization. While the idea makes intuitive sense, theoretical analysis of its effectiveness has been lacking. Here theoretical support is provided for one of its conjectured properties, namely, the ability to allow gradient descent to succeed with less tuning of learning rates. It is shown that even if we fix the learning rate of scale-invariant parameters (e.g., weights of each layer with BN) to a constant (say, $0.3$), gradient descent still approaches a stationary point (i.e., a solution where gradient is zero) in the rate of $T^{-1/2}$ in $T$ iterations, asymptotically matching the best bound for gradient descent with well-tuned learning rates. A similar result with convergence rate $T^{-1/4}$ is also shown for stochastic gradient descent.

## 1 Introduction

Batch Normalization (abbreviated as BatchNorm or BN) (Ioffe & Szegedy, 2015) is one of the most important innovation in deep learning, widely used in modern neural network architectures such as ResNet (He et al., 2016), Inception (Szegedy et al., 2017), and DenseNet (Huang et al., 2017). It also inspired a series of other normalization methods (Ulyanov et al., 2016; Ba et al., 2016; Ioffe, 2017; Wu & He, 2018).

BatchNorm consists of standardizing the output of each layer to have zero mean and unit variance. For a single neuron, if $x_1, \ldots, x_B$ is the original outputs in a mini-batch, then it adds a BatchNorm layer which modifies the outputs to

$$\mathrm{BN}(x_i) = \gamma \frac{x_i - \mu}{\sigma} + \beta, \tag{1}$$

where $\mu = \frac{1}{B} \sum_{i=1}^{B} x_i$ and $\sigma^2 = \frac{1}{B} \sum_{i=1}^{B} (x_i - \mu)^2$ are the mean and variance within the mini-batch, and $\gamma, \beta$ are two learnable parameters. BN appears to stabilize and speed up training, and improve generalization. The inventors suggested (Ioffe & Szegedy, 2015) that these benefits derive from the following:

1. By stabilizing layer outputs it reduces a phenomenon called *Internal Covariate Shift*, whereby the training of a higher layer is continuously undermined or undone by changes in the distribution of its inputs due to parameter changes in previous layers.,

2. Making the weights invariant to scaling, appears to reduce the dependence of training on the scale of parameters and enables us to use a higher learning rate;

3. By implictly regularizing the model it improves generalization.

---

*Work done while visiting Princeton University.

But these three benefits are not fully understood in theory. Understanding generalization for deep models remains an open problem (with or without BN). Furthermore, in demonstration that intuition can sometimes mislead, recent experimental results suggest that BN does not reduce internal covariate shift either (Santurkar et al., 2018), and the authors of that study suggest that the true explanation for BN's effectiveness may lie in a smoothening effect (i.e., lowering of the Hessian norm) on the objective. Another recent paper (Kohler et al., 2018) tries to quantify the benefits of BN for simple machine learning problems such as regression but does not analyze deep models.

**Provable quantification of Effect 2 (learning rates).**    Our study consists of quantifying the effect of BN on learning rates. Ioffe & Szegedy (2015) observed that without BatchNorm, a large learning rate leads to a rapid growth of the parameter scale. Introducing BatchNorm usually stabilizes the growth of weights and appears to implicitly tune the learning rate so that the effective learning rate adapts during the course of the algorithm. They explained this intuitively as follows. After BN the output of a neuron $z = \mathrm{BN}(\boldsymbol{w}^\top \boldsymbol{x})$ is unaffected when the weight $\boldsymbol{w}$ is scaled, i.e., for any scalar $c > 0$,
$$\mathrm{BN}(\boldsymbol{w}^\top \boldsymbol{x}) = \mathrm{BN}((c\boldsymbol{w})^\top \boldsymbol{x}).$$
Taking derivatives one finds that the gradient at $c\boldsymbol{w}$ equals to the gradient at $\boldsymbol{w}$ multiplied by a factor $1/c$. Thus, even though the scale of weight parameters of a linear layer proceeding a BatchNorm no longer means anything to the function represented by the neural network, their growth has an effect of reducing the learning rate.

Our paper considers the following question: *Can we rigorously capture the above intuitive behavior?* Theoretical analyses of speed of gradient descent algorithms in nonconvex settings  study the number of iterations required for convergence to a stationary point (i.e., where gradient vanishes). But they need to assume that the learning rate has been set (magically) to a small enough number determined by the smoothness constant of the loss function — which in practice are of course unknown. With this tuned learning rate, the norm of the gradient reduces asymptotically as $T^{-1/2}$ in $T$ iterations. In case of stochastic gradient descent, the reduction is like $T^{-1/4}$. Thus a potential way to quantify the rate-tuning behavior of BN would be to show that even when the learning rate is fixed to a suitable constant, say $0.1$, from the start, after introducing BN the convergence to stationary point is asymptotically just as fast (essentially) as it would be with a hand-tuned learning rate required by earlier analyses. The current paper rigorously establishes such *auto-tuning* behavior of BN (See below for an important clarification about scale-invariance).

We note that a recent paper (Wu et al., 2018) introduced a new algorithm WNgrad that is motivated by BN and provably has the above auto-tuning behavior as well. That paper did not establish such behavior for BN itself, but it was a clear inspiration for our analysis of BN.

**Scale-invariant and scale-variant parameters.**    The intuition of Ioffe & Szegedy (2015) applies for all scale-invariant parameters, but the actual algorithm also involves other parameters such as $\gamma$ and $\beta$ whose scale does matter. Our analysis partitions the parameters in the neural networks into two groups $W$ (*scale-invariant*) and $\boldsymbol{g}$ (*scale-variant*). The first group, $W = \{\boldsymbol{w}^{(1)}, \ldots, \boldsymbol{w}^{(m)}\}$, consists of all the parameters whose scales does not affect the loss, i.e., scaling $\boldsymbol{w}^{(i)}$ to $c\boldsymbol{w}^{(i)}$ for any $c > 0$ does not change the loss (see Definition 2.1 for a formal definition); the second group, $\boldsymbol{g}$, consists of all other parameters that are not scale-invariant. In a feedforward neural network with BN added at each layer, the layer weights are all scale-invariant. This is also true for BN with $\ell_p$ normalization strategies (Santurkar et al., 2018; Hoffer et al., 2018) and other normalization layers, such as Weight Normalization (Salimans & Kingma, 2016), Layer Normalization (Ba et al., 2016), Group Normalization (Wu & He, 2018) (see Table 1 in Ba et al. (2016) for a summary).

## 1.1 OUR CONTRIBUTIONS

In this paper, we show that the scale-invariant parameters do not require rate tuning for lowering the training loss. To illustrate this, we consider the case in which we set learning rates separately for scale-invariant parameters $W$ and scale-variant parameters $\boldsymbol{g}$. Under some assumptions on the smoothness of the loss and the boundedness of the noise, we show that

1.  In full-batch gradient descent, if the learning rate for $\boldsymbol{g}$ is set optimally, then no matter how the learning rates for $W$ is set, $(W; \boldsymbol{g})$ converges to a first-order stationary point in the rate

$O(T^{-1/2})$, which asymptotically matches with the convergence rate of gradient descent with optimal choice of learning rates for all parameters (Theorem 3.1);

2. In stochastic gradient descent, if the learning rate for $\boldsymbol{g}$ is set optimally, then no matter how the learning rate for $W$ is set, $(W; \boldsymbol{g})$ converges to a first-order stationary point in the rate $O(T^{-1/4} \operatorname{polylog}(T))$, which asymptotically matches with the convergence rate of gradient descent with optimal choice of learning rates for all parameters (up to a $\operatorname{polylog}(T)$ factor) (Theorem 4.2).

In the usual case where we set a unified learning rate for all parameters, our results imply that we only need to set a learning rate that is suitable for $\boldsymbol{g}$. This means introducing scale-invariance into neural networks potentially reduces the efforts to tune learning rates, since there are less number of parameters we need to concern in order to guarantee an asymptotically fastest convergence.

In our study, the loss function is assumed to be smooth. However, BN introduces non-smoothness in extreme cases due to division by zero when the input variance is zero (see equation 1). Note that the suggested implementation of BN by Ioffe & Szegedy (2015) uses a smoothening constant in the whitening step, but it does not preserve scale-invariance. In order to avoid this issue, we describe a simple modification of the smoothening that maintains scale-invariance. Also, our result cannot be applied to neural networks with ReLU, but it is applicable for its smooth approximation softplus (Dugas et al., 2001).

We include some experiments in Appendix D, showing that it is indeed the auto-tuning behavior we analysed in this paper empowers BN to have such convergence with arbitrary learning rate for scale-invariant parameters. In the generalization aspect, a tuned learning rate is still needed for the best test accuracy, and we showed in the experiments that the auto-tuning behavior of BN also leads to a wider range of suitable learning rate for good generalization.

## 1.2 RELATED WORKS

**Previous work for understanding Batch Normalization.** Only a few recent works tried to theoretically understand BatchNorm. Santurkar et al. (2018) was described earlier. Kohler et al. (2018) aims to find theoretical setting such that training neural networks with BatchNorm is faster than without BatchNorm. In particular, the authors analyzed three types of shallow neural networks, but rather than consider gradient descent, the authors designed task-specific training methods when discussing neural networks with BatchNorm. Bjorck et al. (2018) observes that the higher learning rates enabled by BatchNorm improves generalization.

**Convergence of adaptive algorithms.** Our analysis is inspired by the proof for WNGrad (Wu et al., 2018), where the author analyzed an adaptive algorithm, WNGrad, motivated by Weight Normalization (Salimans & Kingma, 2016). Other works analyzing the convergence of adaptive methods are (Ward et al., 2018; Li & Orabona, 2018; Zou & Shen, 2018; Zhou et al., 2018).

**Invariance by Batch Normalization.** Cho & Lee (2017) proposed to run riemmanian gradient descent on Grassmann manifold $\mathcal{G}(1, n)$ since the weight matrix is scaling invariant to the loss function. Hoffer et al. (2018) observed that the effective stepsize is proportional to $\frac{\eta_{\mathrm{w}}}{\|\boldsymbol{w}_t\|^2}$.

## 2 GENERAL FRAMEWORK

In this section, we introduce our general framework in order to study the benefits of scale-invariance.

### 2.1 MOTIVATING EXAMPLES OF NEURAL NETWORKS

Scale-invariance is common in neural networks with BatchNorm. We formally state the definition of scale-invariance below:

**Definition 2.1.** (Scale-invariance) Let $\mathcal{F}(\boldsymbol{w}, \boldsymbol{\theta}')$ be a loss function. We say that $\boldsymbol{w}$ is a scale-invariant parameter of $\mathcal{F}$ if for all $c > 0$, $\mathcal{F}(\boldsymbol{w}, \boldsymbol{\theta}') = \mathcal{F}(c\boldsymbol{w}, \boldsymbol{\theta}')$; if $\boldsymbol{w}$ is not scale-invariant, then we say $\boldsymbol{w}$ is a scale-variant parameter of $\mathcal{F}$.

We consider the following $L$-layer "fully-batch-normalized" feedforward network $\Phi$ for illustration:

$$\mathcal{L}(\boldsymbol{\theta}) = \mathbb{E}_{Z \sim \mathcal{D}^B} \left[ \frac{1}{B} \sum_{b=1}^{B} f_{y_b}(\text{BN}(W^{(L)} \sigma(\text{BN}(W^{(L-1)} \cdots \sigma(\text{BN}(W^{(1)} \boldsymbol{x}_b))))))\right]. \qquad (2)$$

$Z = \{(\boldsymbol{x}_1, y_1), \ldots, (\boldsymbol{x}_B, y_B)\}$ is a mini-batch of $B$ pairs of input data and ground-truth label from a data set $\mathcal{D}$. $f_y$ is an objective function depending on the label, e.g., $f_y$ could be a cross-entropy loss in classification tasks. $W^{(1)}, \ldots, W^{(L)}$ are weight matrices of each layer. $\sigma : \mathbb{R} \to \mathbb{R}$ is a nonlinear activation function which processes its input elementwise (such as ReLU, sigmoid). Given a batch of inputs $\boldsymbol{z}_1, \ldots, \boldsymbol{z}_B \in \mathbb{R}^m$, $\text{BN}(\boldsymbol{z}_b)$ outputs a vector $\tilde{\boldsymbol{z}}_b$ defined as

$$\tilde{z}_{b,k} := \gamma_k \frac{z_{b,k} - \mu_k}{\sigma_k} + \beta_k, \qquad (3)$$

where $\mu_k = \mathbb{E}_{b \in [B]}[z_{b,k}]$ and $\sigma_k^2 = \mathbb{E}_{b \in [B]}[(z_{b,k} - \mu_k)^2]$ are the mean and variance of $\boldsymbol{z}_b$, $\gamma_k$ and $\beta_k$ are two learnable parameters which rescale and offset the normalized outputs to retain the representation power. The neural network $\Phi$ is thus parameterized by weight matrices $W^{(i)}$ in each layer and learnable parameters $\gamma_k, \beta_k$ in each BN.

BN has the property that the output is unchanged when the batch inputs $z_{1,k}, \ldots, z_{B,k}$ are scaled or shifted simultaneously. For $z_{b,k} = \boldsymbol{w}_k^\top \hat{\boldsymbol{x}}_b$ being the output of a linear layer, it is easy to see that $\boldsymbol{w}_k$ is scale-invariant, and thus each row vector of weight matrices $W^{(1)}, \ldots, W^{(L)}$ in $\Phi$ are scale-invariant parameters of $\mathcal{L}(\boldsymbol{\theta})$. In convolutional neural networks with BatchNorm, a similar argument can be done. In particular, each filter of convolutional layer normalized by BN is scale-invariant.

With a general nonlinear activation, other parameters in $\Phi$, the scale and shift parameters $\gamma_k$ and $\beta_k$ in each BN, are scale-variant. When ReLU or Leaky ReLU (Maas et al., 2013) are used as the activation $\sigma$, the vector $(\gamma_1, \ldots, \gamma_m, \beta_1, \ldots, \beta_m)$ of each BN at layer $1 \leq i < L$ (except the last one) is indeed scale-invariant. This can be deduced by using the the (positive) homogeneity of these two types of activations and noticing that the output of internal activations is processed by a BN in the next layer. Nevertheless, we are not able to analyse either ReLU or Leaky ReLU activations because we need the loss to be smooth in our analysis. We can instead analyse smooth activations, such as sigmoid, tanh, softplus (Dugas et al., 2001), etc.

## 2.2 FRAMEWORK

Now we introduce our general framework. Let $\Phi$ be a neural network parameterized by $\boldsymbol{\theta}$. Let $\mathcal{D}$ be a dataset, where each data point $\boldsymbol{z} \sim \mathcal{D}$ is associated with a loss function $\mathcal{F}_{\boldsymbol{z}}(\boldsymbol{\theta})$ ($\mathcal{D}$ can be the set of all possible mini-batches). We partition the parameters $\boldsymbol{\theta}$ into $(W; \boldsymbol{g})$, where $W = \{\boldsymbol{w}^{(1)}, \ldots, \boldsymbol{w}^{(m)}\}$ consisting of parameters that are scale-invariant to all $\mathcal{F}_{\boldsymbol{z}}$, and $\boldsymbol{g}$ contains the remaining parameters. The goal of training the neural network is to minimize the expected loss over the dataset: $\mathcal{L}(W; \boldsymbol{g}) := \mathbb{E}_{\boldsymbol{z} \sim \mathcal{D}}[\mathcal{F}_{\boldsymbol{z}}(W; \boldsymbol{g})]$. In order to illustrate the optimization benefits of scale-invariance, we consider the process of training this neural network by stochastic gradient descent with separate learning rates for $W$ and $\boldsymbol{g}$:

$$\boldsymbol{w}_{t+1}^{(i)} \leftarrow \boldsymbol{w}_t^{(i)} - \eta_{\mathrm{w},t} \nabla_{\boldsymbol{w}_t^{(i)}} \mathcal{F}_{\boldsymbol{z}_t}(\boldsymbol{\theta}_t), \qquad \boldsymbol{g}_{t+1} \leftarrow \boldsymbol{g}_t - \eta_{\mathrm{g},t} \nabla_{\boldsymbol{g}_t} \mathcal{F}_{\boldsymbol{z}_t}(\boldsymbol{\theta}_t). \qquad (4)$$

## 2.3 THE INTRINSIC OPTIMIZATION PROBLEM

Thanks to the scale-invariant properties, the scale of each weight $\boldsymbol{w}^{(i)}$ does not affect loss values. However, the scale does affect the gradients. Let $V = \{\boldsymbol{v}^{(1)}, \ldots, \boldsymbol{v}^{(m)}\}$ be the set of normalized weights, where $\boldsymbol{v}^{(i)} = \boldsymbol{w}^{(i)}/\|\boldsymbol{w}^{(i)}\|_2$. The following simple lemma can be easily shown:

**Lemma 2.2** (Implied by Ioffe & Szegedy (2015)). *For any $W$ and $\boldsymbol{g}$,*

$$\nabla_{\boldsymbol{w}^{(i)}} \mathcal{F}_{\boldsymbol{z}}(W; \boldsymbol{g}) = \frac{1}{\|\boldsymbol{w}^{(i)}\|_2} \nabla_{\boldsymbol{v}^{(i)}} \mathcal{F}_{\boldsymbol{z}}(V; \boldsymbol{g}), \qquad \nabla_{\boldsymbol{g}} \mathcal{F}_{\boldsymbol{z}}(W; \boldsymbol{g}) = \nabla_{\boldsymbol{g}} \mathcal{F}_{\boldsymbol{z}}(V; \boldsymbol{g}). \qquad (5)$$

To make $\|\nabla_{\boldsymbol{w}^{(i)}} \mathcal{F}_{\boldsymbol{z}}(W; \boldsymbol{g})\|_2$ to be small, one can just scale the weights by a large factor. Thus there are ways to reduce the norm of the gradient that do not reduce the loss.

For this reason, we define the intrinsic optimization problem for training the neural network. Instead of optimizing $W$ and $\boldsymbol{g}$ over all possible solutions, we focus on parameters $\boldsymbol{\theta}$ in which $\|\boldsymbol{w}^{(i)}\|_2 = 1$ for all $\boldsymbol{w}^{(i)} \in W$. This does not change our objective, since the scale of $W$ does not affect the loss.

**Definition 2.3** (Intrinsic optimization problem). Let $\mathcal{U} = \{\boldsymbol{\theta} \mid \|\boldsymbol{w}^{(i)}\|_2 = 1 \text{ for all } i\}$ be the *intrinsic domain*. The *intrinsic optimization problem* is defined as optimizing the original problem in $\mathcal{U}$:

$$\min_{(W;\boldsymbol{g})\in\mathcal{U}} \mathcal{L}(W;\boldsymbol{g}). \tag{6}$$

For $\{\boldsymbol{\theta}_t\}$ being a sequence of points for optimizing the original optimization problem, we can define $\{\tilde{\boldsymbol{\theta}}_t\}$, where $\tilde{\boldsymbol{\theta}}_t = (V_t; \boldsymbol{g}_t)$, as a sequence of points optimizing the intrinsic optimization problem.

In this paper, we aim to show that training neural network for the original optimization problem by gradient descent can be seen as training by adaptive methods for the intrinsic optimization problem, and it converges to a first-order stationary point in the intrinsic optimization problem with no need for tuning learning rates for $W$.

## 2.4 ASSUMPTIONS ON THE LOSS

We assume $\mathcal{F}_{\boldsymbol{z}}(W; \boldsymbol{g})$ is defined and twice continuously differentiable at any $\boldsymbol{\theta}$ satisfying none of $\boldsymbol{w}^{(i)}$ is 0. Also, we assume that the expected loss $\mathcal{L}(\boldsymbol{\theta})$ is lower-bounded by $\mathcal{L}_{\min}$.

Furthermore, for $V = \{\boldsymbol{v}^{(1)}, \ldots, \boldsymbol{v}^{(m)}\}$, where $\boldsymbol{v}^{(i)} = \boldsymbol{w}^{(i)}/\|\boldsymbol{w}^{(i)}\|_2$, we assume that the following bounds on the smoothness:

$$\left\|\frac{\partial}{\partial\boldsymbol{v}^{(i)}\partial\boldsymbol{v}^{(j)}}\mathcal{F}_{\boldsymbol{z}}(V;\boldsymbol{g})\right\|_2 \le L_{ij}^{\mathrm{vv}}, \qquad \left\|\frac{\partial}{\partial\boldsymbol{v}^{(i)}\partial\boldsymbol{g}}\mathcal{F}_{\boldsymbol{z}}(V;\boldsymbol{g})\right\|_2 \le L_i^{\mathrm{vg}}, \qquad \left\|\nabla_{\boldsymbol{g}}^2\mathcal{F}_{\boldsymbol{z}}(V;\boldsymbol{g})\right\|_2 \le L^{\mathrm{gg}}.$$

In addition, we assume that the noise on the gradient of $\boldsymbol{g}$ in SGD is upper bounded by $G_{\mathrm{g}}$:

$$\mathbb{E}\left[\|\nabla_{\boldsymbol{g}}\mathcal{F}_{\boldsymbol{z}}(V;\boldsymbol{g}) - \mathbb{E}_{\boldsymbol{z}\sim\mathcal{D}}\left[\nabla_{\boldsymbol{g}}\mathcal{F}_{\boldsymbol{z}}(V;\boldsymbol{g})\right]\|_2^2\right] \le G_{\mathrm{g}}^2.$$

**Smoothed version of motivating neural networks.** Note that the neural network $\Phi$ illustrated in Section 2.1 does not meet the conditions of the smooothness at all since the loss function could be non-smooth. We can make some mild modifications to the motivating example to smoothen it [1]:

(1). The activation could be non-smooth. A possible solution is to use smooth nonlinearities, e.g., sigmoid, tanh, softplus (Dugas et al., 2001), etc. Note that softplus can be seen as a smooth approximation of the most commonly used activation ReLU.

(2). The formula of BN shown in equation 3 may suffer from the problem of division by zero. To avoid this, the inventors of BN, Ioffe & Szegedy (2015), add a small smoothening parameter $\epsilon > 0$ to the denominator, i.e.,

$$\tilde{z}_{b,k} := \gamma_k\frac{z_{b,k} - \mu_k}{\sqrt{\sigma_k^2 + \epsilon}} + \beta_k, \tag{7}$$

However, when $z_{b,k} = \boldsymbol{w}_k^\top\hat{\boldsymbol{x}}_b$, adding a constant $\epsilon$ directly breaks the scale-invariance of $\boldsymbol{w}_k$. We can preserve the scale-invariance by making the smoothening term propositional to $\|\boldsymbol{w}_k\|_2$, i.e., replacing $\epsilon$ with $\epsilon\|\boldsymbol{w}_k\|_2$. By simple linear algebra and letting $\boldsymbol{u} := \mathbb{E}_{b\in[B]}[\hat{\boldsymbol{x}}_b]$, $\boldsymbol{S} := \mathrm{Var}_{b\in[B]}(\hat{\boldsymbol{x}}_b)$, this smoothed version of BN can also be written as

$$\tilde{z}_{b,k} := \gamma_k\frac{\boldsymbol{w}_k^\top(\hat{\boldsymbol{x}}_b - \boldsymbol{u})}{\|\boldsymbol{w}_k\|_{\boldsymbol{S}+\epsilon\boldsymbol{I}}} + \beta_k. \tag{8}$$

Since the variance of inputs is usually large in practice, for small $\epsilon$, the effect of the smoothening term is negligible except in extreme cases.

Using the above two modifications, the loss function is already smooth. However, the scale of scale-variant parameters may be unbounded during training, which could cause the smoothness unbounded. To avoid this issue, we can either project scale-variant parameters to a bounded set, or use weight decay for those parameters (see Appendix C for a proof for the latter solution).

---

[1]Our results to this network are rather conceptual, since the smoothness upper bound can be as large as $M^{O(L)}$, where $L$ is the number of layers and $M$ is the maximum width of each layer.

## 2.5 Key observation: the growth of weights

The following lemma is our key observation. It establishes a connection between the scale-invariant property and the growth of weight scale, which further implies an automatic decay of learning rates:

**Lemma 2.4.** *For any scale-invariant weight $\boldsymbol{w}^{(i)}$ in the network $\Psi$, we have:*

(1). $\boldsymbol{w}_t^{(i)}$ *and* $\nabla_{\boldsymbol{w}_t^{(i)}}\mathcal{F}_{\boldsymbol{z}_t}(\boldsymbol{\theta}_t)$ *are always perpendicular;*

(2). $\|\boldsymbol{w}_{t+1}^{(i)}\|_2^2 = \|\boldsymbol{w}_t^{(i)}\|_2^2 + \eta_{\mathrm{w},t}^2 \|\nabla_{\boldsymbol{w}_t^{(i)}}\mathcal{F}_{\boldsymbol{z}_t}(\boldsymbol{\theta}_t)\|_2^2 = \|\boldsymbol{w}_t^{(i)}\|_2^2 + \frac{\eta_{\mathrm{w},t}^2}{\|\boldsymbol{w}_t^{(i)}\|_2^2}\|\nabla_{\boldsymbol{v}_t^{(i)}}\mathcal{F}_{\boldsymbol{z}_t}(\tilde{\boldsymbol{\theta}}_t)\|_2^2.$

*Proof.* Let $\boldsymbol{\theta}_t'$ be all the parameters in $\boldsymbol{\theta}_t$ other than $\boldsymbol{w}_t^{(i)}$. Taking derivatives with respect to $c$ for the both sides of $\mathcal{F}_{\boldsymbol{z}_t}(\boldsymbol{w}_t^{(i)}, \boldsymbol{\theta}_t') = \mathcal{F}_{\boldsymbol{z}_t}(c\boldsymbol{w}_t^{(i)}, \boldsymbol{\theta}_t')$, we have $0 = \frac{\partial}{\partial c}\mathcal{F}_{\boldsymbol{z}_t}(c\boldsymbol{w}_t^{(i)}, \boldsymbol{\theta}_t')$. The right hand side equals $\nabla_{c\boldsymbol{w}_t^{(i)}}\mathcal{F}_{\boldsymbol{z}_t}(c\boldsymbol{w}_t^{(i)}, \boldsymbol{\theta}_t')^\top \boldsymbol{w}_t^{(i)}$, so the first proposition follows by taking $c = 1$. Applying Pythagorean theorem and Lemma 2.2, the second proposition directly follows. $\qquad\square$

Using Lemma 2.4, we can show that performing gradient descent for the original problem is equivalent to performing an adaptive gradient method for the intrinsic optimization problem:

**Theorem 2.5.** *Let $G_t^{(i)} = \|\boldsymbol{w}_t^{(i)}\|_2^2$. Then for all $t \geq 0$,*

$$\boldsymbol{v}_{t+1}^{(i)} = \Pi\left(\boldsymbol{v}_t^{(i)} - \frac{\eta_{\mathrm{w},t}}{G_t^{(i)}}\nabla_{\boldsymbol{v}_t^{(i)}}\mathcal{F}_{\boldsymbol{z}_t}(\tilde{\boldsymbol{\theta}}_t)\right), \qquad G_{t+1}^{(i)} = G_t^{(i)} + \frac{\eta_{\mathrm{w},t}^2}{G_t^{(i)}}\|\nabla_{\boldsymbol{v}_t^{(i)}}\mathcal{F}_{\boldsymbol{z}_t}(\tilde{\boldsymbol{\theta}}_t)\|_2^2, \quad (9)$$

*where $\Pi$ is a projection operator which maps any vector $\boldsymbol{w}$ to $\boldsymbol{w}/\|\boldsymbol{w}\|_2$.*

**Remark 2.6.** *Wu et al. (2018) noticed that Theorem 2.5 is true for Weight Normalization by direct calculation of gradients. Inspiring by this, they proposed a new adaptive method called* WNGrad. *Our theorem is more general since it holds for any normalization methods as long as it induces scale-invariant properties to the network. The adaptive update rule derived in our theorem can be seen as* WNGrad *with projection to unit sphere after each step.*

*Proof for Theorem 2.5.* Using Lemma 2.2, we have

$$\boldsymbol{w}_{t+1}^{(i)} = \boldsymbol{w}_t^{(i)} - \frac{\eta_{\mathrm{w},t}}{\|\boldsymbol{w}_t^{(i)}\|_2}\nabla_{\boldsymbol{v}_t^{(i)}}\mathcal{F}_{\boldsymbol{z}_t}(\tilde{\boldsymbol{\theta}}_t) = \|\boldsymbol{w}_t^{(i)}\|_2\left(\boldsymbol{v}_t^{(i)} - \frac{\eta_{\mathrm{w},t}}{G_t^{(i)}}\nabla_{\boldsymbol{v}_t^{(i)}}\mathcal{F}_{\boldsymbol{z}_t}(\tilde{\boldsymbol{\theta}}_t)\right),$$

which implies the first equation. The second equation is by Lemma 2.4. $\qquad\square$

While popular adaptive gradient methods such as AdaGrad (Duchi et al., 2011), RMSprop (Tieleman & Hinton, 2012), Adam (Kingma & Ba, 2014) adjust learning rates for each single coordinate, the adaptive gradient method described in Theorem 2.5 sets learning rates $\eta_{\mathrm{w},t}/G_t^{(i)}$ for each scale-invariant parameter respectively. In this paper, we call $\eta_{\mathrm{w},t}/G_t^{(i)}$ the *effective learning rate* of $\boldsymbol{v}_t^{(i)}$ or $\boldsymbol{w}_t^{(i)}$, because it is $\eta_{\mathrm{w},t}/\|\boldsymbol{w}_t^{(i)}\|_2^2$ instead of $\eta_{\mathrm{w},t}$ alone that really determines the trajectory of gradient descent given the normalized scale-invariant parameter $\boldsymbol{v}_t^{(i)}$. In other words, the magnitude of the initialization of parameters before BN is as important as their learning rates: multiplying the initialization of scale-invariant parameter by constant $C$ is equivalent to dividing its learning rate by $C^2$. Thus we suggest researchers to report initialization as well as learning rates in the future experiments.

## 3 Training by full-batch gradient descent

In this section, we rigorously analyze the effect related to the scale-invariant properties in training neural network by full-batch gradient descent. We use the framework introduced in Section 2.2 and assumptions from Section 2.4. We focus on the full-batch training, i.e., $\boldsymbol{z}_t$ is always equal to the whole training set and $\mathcal{F}_{\boldsymbol{z}_t}(\boldsymbol{\theta}) = \mathcal{L}(\boldsymbol{\theta})$.

### 3.1 SETTINGS AND MAIN THEOREM

**Assumptions on learning rates.** We consider the case that we use fixed learning rates for both $W$ and $g$, i.e., $\eta_{\mathrm{w},0} = \cdots = \eta_{\mathrm{w},T-1} = \eta_{\mathrm{w}}$ and $\eta_{\mathrm{g},0} = \cdots = \eta_{\mathrm{g},T-1} = \eta_{\mathrm{g}}$. We assume that $\eta_{\mathrm{g}}$ is tuned carefully to $\eta_{\mathrm{g}} = (1 - c_{\mathrm{g}})/L^{\mathrm{gg}}$ for some constant $c_{\mathrm{g}} \in (0,1)$. For $\eta_{\mathrm{w}}$, we do not make any assumption, i.e., $\eta_{\mathrm{w}}$ can be set to any positive value.

**Theorem 3.1.** *Consider the process of training $\Phi$ by gradient descent with $\eta_{\mathrm{g}} = 2(1-c_{\mathrm{g}})/L^{\mathrm{gg}}$ and arbitrary $\eta_{\mathrm{w}} > 0$. Then $\Phi$ converges to a stationary point in the rate of*

$$\min_{0 \le t < T} \|\nabla \mathcal{L}(V_t; \boldsymbol{g}_t)\|_2 \le \tilde{O}\left(\frac{1}{\sqrt{T}}\left(\frac{1}{\sqrt{\eta_{\mathrm{w}}}} + \eta_{\mathrm{w}}^2 + \frac{1+\eta_{\mathrm{w}}}{\sqrt{\eta_{\mathrm{g}}}}\right)\right), \tag{10}$$

*where $V_t = \{\boldsymbol{v}_t^{(1)}, \ldots, \boldsymbol{v}_t^{(m)}\}$ with $\boldsymbol{v}_t^{(i)} = \boldsymbol{w}_t^{(i)}/\|\boldsymbol{w}_t^{(i)}\|_2$, $\tilde{O}$ suppresses polynomial factors in $L_{ij}^{\mathrm{vv}}$, $L_i^{\mathrm{vg}}$, $L^{\mathrm{gg}}$, $\|\boldsymbol{w}_0^{(i)}\|_2$, $\|\boldsymbol{w}_0^{(i)}\|_2^{-1}$, $\mathcal{L}(\boldsymbol{\theta}_0) - \mathcal{L}_{\min}$ for all $i, j$, and we see $L^{\mathrm{gg}} = \Omega(1)$.*

This matches the asymptotic convergence rate of GD by Carmon et al. (2018).

### 3.2 PROOF SKETCH

The high level idea is to use the decrement of loss function to upper bound the sum of the squared norm of the gradients. Note that $\|\nabla \mathcal{L}(V_t; \boldsymbol{g}_t)\|_2^2 = \sum_{i=1}^m \|\nabla_{\boldsymbol{v}_t^{(i)}} \mathcal{L}(V_t; \boldsymbol{g}_t)\|_2^2 + \|\nabla_{\boldsymbol{g}_t} \mathcal{L}(V_t; \boldsymbol{g}_t)\|_2^2$. For the first part $\sum_{i=1}^m \|\nabla_{\boldsymbol{v}_t^{(i)}} \mathcal{L}(V_t; \boldsymbol{g}_t)\|_2^2$, we have

$$\begin{aligned}
\sum_{i=1}^m \sum_{t=0}^{T-1} \|\nabla_{\boldsymbol{v}_t^{(i)}} \mathcal{L}(V_t; \boldsymbol{g}_t)\|_2^2 &= \sum_{i=1}^m \sum_{t=0}^{T-1} \|\boldsymbol{w}_t^{(i)}\|_2^2 \|\nabla_{\boldsymbol{w}_t^{(i)}} \mathcal{L}(W_t; \boldsymbol{g}_t)\|_2^2 \\
&\le \sum_{i=1}^m \|\boldsymbol{w}_T^{(i)}\|_2^2 \cdot \frac{\|\boldsymbol{w}_T^{(i)}\|_2^2 - \|\boldsymbol{w}_0^{(i)}\|_2^2}{\eta_{\mathrm{w}}^2}
\end{aligned} \tag{11}$$

Thus the core of the proof is to show that the monotone increasing $\|\boldsymbol{w}_T^{(i)}\|_2$ has an upper bound for all $T$. It is shown that for every $\boldsymbol{w}^{(i)}$, the whole training process can be divided into at most two phases. In the first phase, the effective learning rate $\eta_w/\|\boldsymbol{w}_t^{(i)}\|_2^2$ is larger than some threshold $\frac{1}{C_i}$ (defined in Lemma 3.2) and in the second phase it is smaller.

**Lemma 3.2** (Taylor Expansion). *Let $C_i := \frac{1}{2}\sum_{j=1}^m L_{ij}^{\mathrm{vv}} + L_i^{\mathrm{vg}^2} m/(c_{\mathrm{g}} L^{\mathrm{gg}}) = \tilde{O}(1)$. Then*

$$\mathcal{L}(\boldsymbol{\theta}_{t+1}) - \mathcal{L}(\boldsymbol{\theta}_t) = -\sum_{i=1}^m \eta_{\mathrm{w}} \|\nabla_{\boldsymbol{w}_t^{(i)}} \mathcal{L}(\boldsymbol{\theta}_t)\|_2^2 \left(1 - \frac{C_i \eta_{\mathrm{w}}}{\|\boldsymbol{w}_t^{(i)}\|_2^2}\right) - \frac{1}{2}c_{\mathrm{g}}\eta_{\mathrm{g}} \|\nabla_{\boldsymbol{g}_t} \mathcal{L}(\boldsymbol{\theta}_t)\|_2^2. \tag{12}$$

If $\|\boldsymbol{w}_t^{(i)}\|_2$ is large enough and that the process enters the second phase, then by Lemma 3.2 in each step the loss function $\mathcal{L}$ will decrease by $\frac{\eta_{\mathrm{w}}}{2}\|\nabla_{\boldsymbol{w}_t^{(i)}} \mathcal{L}(\boldsymbol{\theta}_t)\|_2^2 = \frac{\|\boldsymbol{w}_{t+1}^{(i)}\|_2^2 - \|\boldsymbol{w}_t^{(i)}\|_2^2}{2\eta_{\mathrm{w}}}$ (Recall that $\|\boldsymbol{w}_{t+1}^{(i)}\|_2^2 - \|\boldsymbol{w}_t^{(i)}\|_2^2 = \eta_{\mathrm{w},t}^2 \|\nabla_{\boldsymbol{w}_t^{(i)}} \mathcal{L}(\boldsymbol{\theta}_t)\|_2^2$ by Lemma 2.4). Since $\mathcal{L}$ is lower-bounded, we can conclude $\|\boldsymbol{w}_t^{(i)}\|_2^2$ is also bounded .

For the second part, we can also show that by Lemma 3.2

$$\sum_{t=0}^{T-1} \|\nabla_{\boldsymbol{g}_t} \mathcal{L}(V_t; \boldsymbol{g}_t)\|_2^2 \le \frac{2}{\eta_{\mathrm{g}} c_{\mathrm{g}}} \left(\mathcal{L}(\boldsymbol{\theta}_0) - \mathcal{L}(\boldsymbol{\theta}_T) + \sum_{i=1}^m \frac{C_i \|\boldsymbol{w}_T^{(i)}\|_2^2}{\|\boldsymbol{w}_0^{(i)}\|_2^2}\right)$$

Thus we can conclude $\tilde{O}(\frac{1}{\sqrt{T}})$ convergence rate of $\|\nabla \mathcal{L}(\boldsymbol{\theta}_t)\|_2$ as follows.

$$\min_{0 \le t < T} \|\nabla \mathcal{L}(V_t; \boldsymbol{g}_t)\|_2^2 \le \frac{1}{T}\sum_{t=0}^{T-1}\left(\sum_{i=1}^m \|\nabla_{\boldsymbol{v}_t^{(i)}} \mathcal{L}(V_t; \boldsymbol{g}_t)\|_2^2 + \sum_{i=1}^m \|\nabla_{\boldsymbol{g}_t} \mathcal{L}(V_t; \boldsymbol{g}_t)\|_2^2\right) \le \tilde{O}(\frac{1}{T})$$

The full proof is postponed to Appendix A.

## 4 TRAINING BY STOCHASTIC GRADIENT DESCENT

In this section, we analyze the effect related to the scale-invariant properties when training a neural network by stochastic gradient descent. We use the framework introduced in Section 2.2 and assumptions from Section 2.4.

### 4.1 SETTINGS AND MAIN THEOREM

**Assumptions on learning rates.** As usual, we assume that the learning rate for $g$ is chosen carefully and the learning rate for $W$ is chosen rather arbitrarily. More specifically, we consider the case that the learning rates are chosen as

$$\eta_{\mathrm{w},t} = \eta_{\mathrm{w}} \cdot (t+1)^{-\alpha}, \qquad \eta_{\mathrm{g},t} = \eta_{\mathrm{g}} \cdot (t+1)^{-1/2}.$$

We assume that the initial learning rate $\eta_{\mathrm{g}}$ of $g$ is tuned carefully to $\eta_{\mathrm{g}} = (1 - c_{\mathrm{g}})/L^{\mathrm{gg}}$ for some constant $c_{\mathrm{g}} \in (0,1)$. Note that this learning rate schedule matches the best known convergence rate $O(T^{-1/4})$ of SGD in the case of smooth non-convex loss functions (Ghadimi & Lan, 2013).

For the learning rates of $W$, we only assume that $0 \le \alpha \le 1/2$, i.e., the learning rate decays equally as or slower than the optimal SGD learning rate schedule. $\eta_{\mathrm{w}}$ can be set to any positive value. Note that this includes the case that we set a fixed learning rate $\eta_{\mathrm{w},0} = \cdots = \eta_{\mathrm{w},T-1} = \eta_{\mathrm{w}}$ for $W$ by taking $\alpha = 0$.

**Remark 4.1.** *Note that the auto-tuning behavior induced by scale-invariances always decreases the learning rates. Thus, if we set $\alpha > 1/2$, there is no hope to adjust the learning rate to the optimal strategy $\Theta(t^{-1/2})$. Indeed, in this case, the learning rate $1/G_t$ in the intrinsic optimization process decays exactly in the rate of $\tilde{\Theta}(t^{-\alpha})$, which is the best possible learning rate can be achieved without increasing the original learning rate.*

**Theorem 4.2.** *Consider the process of training $\Phi$ by gradient descent with $\eta_{\mathrm{w},t} = \eta_{\mathrm{w}} \cdot (t+1)^{-\alpha}$ and $\eta_{\mathrm{g},t} = \eta_{\mathrm{g}} \cdot (t+1)^{-1/2}$, where $\eta_{\mathrm{g}} = 2(1 - c_{\mathrm{g}})/L^{\mathrm{gg}}$ and $\eta_{\mathrm{w}} > 0$ is arbitrary. Then $\Phi$ converges to a stationary point in the rate of*

$$\min_{0 \le t < T} \mathbb{E}\left[\|\nabla \mathcal{L}(V_t; \boldsymbol{g}_t)\|_2^2\right] \le \begin{cases} \tilde{O}\left(\frac{\log T}{\sqrt{T}}\right) & 0 \le \alpha < 1/2; \\ \tilde{O}\left(\frac{(\log T)^{3/2}}{\sqrt{T}}\right) & \alpha = 1/2. \end{cases} \tag{13}$$

*where $V_t = \{\boldsymbol{v}_t^{(1)}, \ldots, \boldsymbol{v}_t^{(m)}\}$ with $\boldsymbol{v}_t^{(i)} = \boldsymbol{w}_t^{(i)}/\|\boldsymbol{w}_t^{(i)}\|_2$, $\tilde{O}$ suppresses polynomial factors in $\eta_{\mathrm{w}}, \eta_{\mathrm{w}}^{-1}, \eta_{\mathrm{g}}^{-1}, L_{ij}^{\mathrm{vv}}, L_i^{\mathrm{vg}}, L^{\mathrm{gg}}, \|\boldsymbol{w}_0^{(i)}\|_2, \|\boldsymbol{w}_0^{(i)}\|_2^{-1}, \mathcal{L}(\theta_0) - \mathcal{L}_{\min}$ for all $i, j$, and we see $L^{\mathrm{gg}} = \Omega(1)$.*

Note that this matches the asymptotic convergence rate of SGD, within a $\mathrm{polylog}(T)$ factor.

### 4.2 PROOF SKETCH

We delay the full proof into Appendix B and give a proof sketch in a simplified setting where there is no $g$ and $\alpha \in [0, \frac{1}{2})$. We also assume there's only one $\boldsymbol{w}_i$, that is, $m = 1$ and omit the index $i$.

By Taylor expansion, we have

$$\mathbb{E}\left[\mathcal{L}(\boldsymbol{\theta}_{t+1})\right] \le \mathcal{L}(\boldsymbol{\theta}_t) - \frac{\eta_{\mathrm{w},t}}{\|\boldsymbol{v}_t\|^2}\|\nabla_{\boldsymbol{w}_t}\mathcal{L}(\boldsymbol{\theta}_t)\|_2^2 + \mathbb{E}\left[\frac{\eta_{\mathrm{w},t}^2 L^{\mathrm{vv}}}{\|\boldsymbol{w}_t\|_2^2}\|\nabla_{\boldsymbol{w}_t}\mathcal{F}_{\boldsymbol{z}_t}(\boldsymbol{\theta}_t)\|_2^2\right] \tag{14}$$

We can lower bound the effective learning rate $\frac{\eta_{\mathrm{w},T}}{\|\boldsymbol{w}_T\|^2}$ and upper bound the second order term respectively in the following way:

(1). For all $0 \le \alpha < \frac{1}{2}$, the effective learning rate $\frac{\eta_{\mathrm{w},T}}{\|\boldsymbol{w}_T\|^2} = \tilde{\Omega}(T^{-1/2})$;

(2). $\sum_{t=0}^{T} \frac{\eta_{\mathrm{w},t}^2 L^{\mathrm{vv}}}{\|\boldsymbol{w}_t\|_2^2}\|\nabla_{\boldsymbol{w}_t}\mathcal{F}_{\boldsymbol{z}_t}(\boldsymbol{\theta}_t)\|_2^2 = \tilde{O}\left(\log\left(\frac{\|\boldsymbol{w}_T^2\|}{\|\boldsymbol{w}_0^2\|}\right)\right) = \tilde{O}(\log T)$.

Taking expectation over equation 14 and summing it up, we have

$$\mathbb{E}\left[\sum_{t=0}^{T-1} \frac{\eta_{\mathrm{w},t}}{\|\boldsymbol{w}_t\|^2}\|\nabla_{\boldsymbol{v}_t}\mathcal{L}(\boldsymbol{\theta}_t)\|_2^2\right] \le \mathcal{L}(\boldsymbol{\theta}_0) - \mathbb{E}[\mathcal{L}(\boldsymbol{\theta}_T)] + \mathbb{E}\left[\sum_{t=0}^{T-1} \frac{\eta_{\mathrm{w},t}^2 L^{\mathrm{vv}}}{\|\boldsymbol{w}_t\|_2^2}\|\nabla_{\boldsymbol{w}_t}\mathcal{F}_{\boldsymbol{z}_t}(\boldsymbol{\theta}_t)\|_2^2\right].$$

Plug the above bounds into the above inequality, we complete the proof.

$$\Omega(T^{-1/2}) \cdot \mathbb{E}\left[\sum_{t=0}^{T-1} \|\nabla_{\boldsymbol{v}_t}\mathcal{L}(\boldsymbol{\theta}_t)\|_2^2\right] \le \mathcal{L}(\boldsymbol{\theta}_0) - \mathbb{E}[\mathcal{L}(\boldsymbol{\theta}_T)] + \tilde{O}(\log T).$$

## 5 CONCLUSIONS AND FUTURE WORKS

In this paper, we studied how scale-invariance in neural networks with BN helps optimization, and showed that (stochastic) gradient descent can achieve the asymptotic best convergence rate without tuning learning rates for scale-invariant parameters. Our analysis suggests that scale-invariance in nerual networks introduced by BN reduces the efforts for tuning learning rate to fit the training data.

However, our analysis only applies to smooth loss functions. In modern neural networks, ReLU or Leaky ReLU are often used, which makes the loss non-smooth. It would have more implications by showing similar results in non-smooth settings. Also, we only considered gradient descent in this paper. It can be shown that if we perform (stochastic) gradient descent with momentum, the norm of scale-invariant parameters will also be monotone increasing. It would be interesting to use it to show similar convergence results for more gradient methods.

### ACKNOWLEDGMENTS

Thanks Yuanzhi Li, Wei Hu and Noah Golowich for helpful discussions. This research was done with support from NSF, ONR, Simons Foundation, Mozilla Research, Schmidt Foundation, DARPA, Amazon, and SRC. We thank Amazon Web Services for providing compute time for the experiments in this paper.

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

## A   PROOF FOR FULL-BATCH GRADIENT DESCENT

By the scale-invariant property of $\boldsymbol{w}^{(i)}$, we know that $\mathcal{L}(W; \boldsymbol{g}) = \mathcal{L}(V; \boldsymbol{g})$. Also, the following identities about derivatives can be easily obtained:

$$\frac{\partial}{\partial \boldsymbol{w}^{(i)} \partial \boldsymbol{w}^{(j)}} \mathcal{L}(W; \boldsymbol{g}) = \frac{1}{\|\boldsymbol{w}^{(i)}\|_2 \|\boldsymbol{w}^{(j)}\|_2} \frac{\partial}{\partial \boldsymbol{v}^{(i)} \partial \boldsymbol{v}^{(j)}} \mathcal{L}(V; \boldsymbol{g})$$

$$\frac{\partial}{\partial \boldsymbol{w}^{(i)} \partial \boldsymbol{g}} \mathcal{L}(W; \boldsymbol{g}) = \frac{1}{\|\boldsymbol{w}^{(i)}\|_2 \|\boldsymbol{w}^{(j)}\|_2} \frac{\partial}{\partial \boldsymbol{v}^{(i)} \partial \boldsymbol{g}} \mathcal{L}(V; \boldsymbol{g})$$

$$\nabla_{\boldsymbol{g}}^2 \mathcal{L}(W; \boldsymbol{g}) = \nabla_{\boldsymbol{g}}^2 \mathcal{L}(V; \boldsymbol{g}).$$

Thus, the assumptions on the smoothness imply

$$\left\| \frac{\partial}{\partial \boldsymbol{w}^{(i)} \partial \boldsymbol{w}^{(j)}} \mathcal{L}(W; \boldsymbol{g}) \right\|_2 \leq \frac{L_{ij}^{\mathrm{vv}}}{\|\boldsymbol{w}^{(i)}\|_2 \|\boldsymbol{w}^{(j)}\|_2} \tag{15}$$

$$\left\| \frac{\partial}{\partial \boldsymbol{w}^{(i)} \partial \boldsymbol{g}} \mathcal{L}(W; \boldsymbol{g}) \right\|_2 \leq \frac{L_i^{\mathrm{vg}}}{\|\boldsymbol{w}^{(i)}\|_2} \tag{16}$$

$$\left\| \nabla_{\boldsymbol{g}}^2 \mathcal{L}(W; \boldsymbol{g}) \right\|_2 \leq L^{\mathrm{gg}}. \tag{17}$$

*Proof for Lemma 3.2.* Using Taylor expansion, we have $\exists \gamma \in (0,1)$, such that for $\boldsymbol{w}_t^{(i')} = (1 - \gamma) \boldsymbol{w}_t^{(i)} + \gamma \boldsymbol{w}_{t+1}^{(i)}$,

$$\begin{aligned}
\mathcal{L}(\theta_{t+1}) - \mathcal{L}(\theta_t) \leq & \sum_{i=1}^m \Delta \boldsymbol{w}_t^{(i)\top} \nabla_{\boldsymbol{w}_t^{(i)}} \mathcal{L}(\theta_t) + \Delta \boldsymbol{g}_t^\top \nabla_{\boldsymbol{g}_t} \mathcal{L}(\theta_t) \\
& + \frac{1}{2} \sum_{i=1}^m \sum_{j=1}^m \left\| \Delta \boldsymbol{w}_t^{(i)} \right\|_2 \left\| \Delta \boldsymbol{w}_t^{(j)} \right\|_2 \frac{L_{ij}^{\mathrm{vv}}}{\left\| \boldsymbol{w}_t^{(i')} \right\|_2 \left\| \boldsymbol{w}_t^{(j')} \right\|_2} \\
& + \sum_{i=1}^m \left\| \Delta \boldsymbol{w}_t^{(i)} \right\|_2 \| \Delta \boldsymbol{g}_t \|_2 \frac{L_i^{\mathrm{vg}}}{\left\| \boldsymbol{w}_t^{(i')} \right\|_2} \\
& + \frac{1}{2} \sum_{i=1}^m \| \Delta \boldsymbol{g}_t \|_2^2 L^{\mathrm{gg}}.
\end{aligned}$$

Note that $\boldsymbol{w}_{t+1}^{(i)} - \boldsymbol{w}_t^{(i)} = \eta_{\mathrm{w}} \nabla_{\boldsymbol{w}_t^{(i)}} \mathcal{L}(\boldsymbol{\theta}_t)$ is perpendicular to $\boldsymbol{w}_t^{(i)}$, we have

$$\| \boldsymbol{w}_t^{(i')} \|_2 \geq \| \gamma (\boldsymbol{w}_{t+1}^{(i)} - \boldsymbol{w}_t^{(i)}) + \boldsymbol{w}_t^{(i)} \|_2 \geq \| \boldsymbol{w}_t^{(i)} \|_2.$$

Thus,

$$\mathcal{L}(\theta_{t+1}) - \mathcal{L}(\theta_t) \le \sum_{i=1}^{m} \Delta \boldsymbol{w}_t^{(i)\top} \nabla_{\boldsymbol{w}_t^{(i)}} \mathcal{L}(\theta_t) + \Delta \boldsymbol{g}_t^{\top} \nabla_{\boldsymbol{g}_t} \mathcal{L}(\theta_t)$$

$$+ \frac{1}{2} \sum_{i=1}^{m} \sum_{j=1}^{m} \left\| \Delta \boldsymbol{w}_t^{(i)} \right\|_2 \left\| \Delta \boldsymbol{w}_t^{(j)} \right\|_2 \frac{L_{ij}^{\mathrm{vv}}}{\left\| \boldsymbol{w}_t^{(i)} \right\|_2 \left\| \boldsymbol{w}_t^{(j)} \right\|_2}$$

$$+ \sum_{i=1}^{m} \left\| \Delta \boldsymbol{w}_t^{(i)} \right\|_2 \|\Delta \boldsymbol{g}_t\|_2 \frac{L_i^{\mathrm{vg}}}{\left\| \boldsymbol{w}_t^{(i)} \right\|_2}$$

$$+ \frac{1}{2} \sum_{i=1}^{m} \|\Delta \boldsymbol{g}_t\|_2^2 L^{\mathrm{gg}}.$$

By the inequality of arithmetic and geometric means, we have

$$\left\| \Delta \boldsymbol{w}_t^{(i)} \right\|_2 \left\| \Delta \boldsymbol{w}_t^{(j)} \right\|_2 \frac{L_{ij}^{\mathrm{vv}}}{\left\| \boldsymbol{w}_t^{(i)} \right\|_2 \left\| \boldsymbol{w}_t^{(j)} \right\|_2} \le \frac{1}{2} \left\| \Delta \boldsymbol{w}_t^{(i)} \right\|_2^2 \frac{L_{ij}^{\mathrm{vv}}}{\left\| \boldsymbol{w}_t^{(i)} \right\|_2^2} + \frac{1}{2} \left\| \Delta \boldsymbol{w}_t^{(j)} \right\|_2^2 \frac{L_{ij}^{\mathrm{vv}}}{\left\| \boldsymbol{w}_t^{(j)} \right\|_2^2}$$

$$\left\| \Delta \boldsymbol{w}_t^{(i)} \right\|_2 \|\Delta \boldsymbol{g}_t\|_2 \frac{L_i^{\mathrm{vg}}}{\left\| \boldsymbol{w}_t^{(i)} \right\|_2} \le \left\| \Delta \boldsymbol{w}_t^{(i)} \right\|_2^2 \frac{L_i^{\mathrm{vg}\,2} m/(c_{\mathrm{g}} L^{\mathrm{gg}})}{\left\| \boldsymbol{w}_t^{(i)} \right\|_2^2} + \frac{1}{4} c_{\mathrm{g}} \|\Delta \boldsymbol{g}_t\|_2^2 \frac{L^{\mathrm{gg}}}{m}.$$

Taking $\Delta \boldsymbol{w}_t^{(i)} = -\eta_{\mathrm{w}} \nabla_{\boldsymbol{w}_t^{(i)}} \mathcal{L}(\theta_t), \Delta \boldsymbol{g}_t = -\eta_{\mathrm{g}} \nabla_{\boldsymbol{g}_t} \mathcal{L}(\theta_t)$, we have

$$\mathcal{L}(\theta_{t+1}) - \mathcal{L}(\theta_t) \le - \sum_{i=1}^{m} \eta_{\mathrm{w}} \|\nabla_{\boldsymbol{w}_t^{(i)}} \mathcal{L}(\theta_t)\|_2^2 - \eta_{\mathrm{g}} \|\nabla_{\boldsymbol{g}_t} \mathcal{L}(\theta_t)\|_2^2$$

$$+ \frac{1}{2} \sum_{i=1}^{m} \left( \sum_{j=1}^{m} L_{ij}^{\mathrm{vv}} + 2 L_i^{\mathrm{vg}\,2} m/(c_{\mathrm{g}} L^{\mathrm{gg}}) \right) \frac{\eta_{\mathrm{w}}^2}{\left\| \boldsymbol{w}_t^{(i)} \right\|_2^2} \|\nabla_{\boldsymbol{w}_t^{(i)}} \mathcal{L}(\theta_t)\|_2^2$$

$$+ \frac{1}{2} (1 + c_{\mathrm{g}}/2) \eta_{\mathrm{g}}^2 \|\nabla_{\boldsymbol{g}_t} \mathcal{L}(\theta_t)\|_2^2 L^{\mathrm{gg}}.$$

We can complete the proof by replacing $\frac{1}{2} \sum_{j=1}^{m} L_{ij}^{\mathrm{vv}} + L_i^{\mathrm{vg}\,2} m/(c_{\mathrm{g}} L^{\mathrm{gg}})$ with $C_i$. $\qquad\square$

Using the assumption on the smoothness, we can show that the gradient with respect to $\boldsymbol{w}^{(i)}$ is essentially bounded:

**Lemma A.1.** *For any $W$ and $\boldsymbol{g}$, we have*

$$\|\nabla_{\boldsymbol{w}^{(i)}} \mathcal{L}(W; \boldsymbol{g})\|_2 \le \frac{\pi L_{ii}^{\mathrm{vv}}}{\|\boldsymbol{w}^{(i)}\|_2}. \tag{18}$$

*Proof.* A.1 Fix all the parameters except $\boldsymbol{w}^{(i)}$. Then $\mathcal{L}(W; \boldsymbol{g})$ can be written as a function $f(\boldsymbol{w}^{(i)})$ on the variable $\boldsymbol{w}^{(i)}$. Let $S = \{\boldsymbol{w}^{(i)} \mid \|\boldsymbol{w}^{(i)}\|_2 = 1\}$. Since $f$ is continuous and $S$ is compact, there must exist $\boldsymbol{v}_{\min}^{(i)} \in S$ such that $f(\boldsymbol{v}_{\min}^{(i)}) \le f(\boldsymbol{w}^{(i)})$ for all $\boldsymbol{w}^{(i)} \in S$. Note that $\boldsymbol{w}^{(i)}$ is scale-invariant, so $\boldsymbol{w}_{\min}^{(i)}$ is also a minimum in the entire domain and $\nabla f(\boldsymbol{w}_{\min}^{(i)}) = 0$.

For an arbitrary $\boldsymbol{w}^{(i)}$, let $\boldsymbol{v}^{(i)} = \boldsymbol{w}^{(i)}/\|\boldsymbol{w}^{(i)}\|_2$. Let $h : [0, 1] \to S$ be a curve such that $h(0) = \boldsymbol{v}_{\min}^{(i)}$, $h(1) = \boldsymbol{v}^{(i)}$, and $h$ goes along the geodesic from $\boldsymbol{v}_{\min}^{(i)}$ to $\boldsymbol{v}^{(i)}$ on the unit sphere $S$ with constant speed. Let $H(\tau) = \nabla f(h(\tau))$. By Taylor expansion, we have

$$\nabla f(\boldsymbol{v}^{(i)}) = H(1) = H(0) + H'(\xi) = \nabla^2 f(h(\xi)) h'(\xi).$$

Thus, $\|\nabla f(\boldsymbol{v}^{(i)})\|_2 \le \pi L_i^{\mathrm{vv}}$ and $\|\nabla_{\boldsymbol{w}^{(i)}} \mathcal{L}(W; \boldsymbol{g})\|_2 \le \frac{\pi L_{ii}^{\mathrm{vv}}}{\|\boldsymbol{w}^{(i)}\|_2}$. $\qquad\square$

The following lemma gives an upper bound to the weight scales.

**Lemma A.2.** *For all $T \geq 0$, we have*

1. *for $i = 1, \ldots, m$, let $t_i$ be the maximum time $0 \leq \tau < T$ satisfying $\|\boldsymbol{w}_\tau^{(i)}\|_2^2 \leq 2C_i \eta_\mathrm{w}$ ($t_i = -1$ if no such $\tau$ exists), then $\|\boldsymbol{w}_{t_i+1}^{(i)}\|_2$ can be bounded by*

$$\|\boldsymbol{w}_{t_i+1}^{(i)}\|_2^2 \leq \|\boldsymbol{w}_0^{(i)}\|_2^2 + \eta_\mathrm{w} \left( 2C_i + (\pi L_i^\mathrm{vv})^2 \frac{\eta_\mathrm{w}}{\|\boldsymbol{w}_0^{(i)}\|_2^2} \right); \qquad (19)$$

2. *the following inequality on weight scales at time $T$ holds:*

$$\sum_{i=1}^m \frac{\|\boldsymbol{w}_T^{(i)}\|_2^2 - \|\boldsymbol{w}_0^{(i)}\|_2^2}{2\eta_\mathrm{w}} + \frac{1}{2} c_\mathrm{g} \eta_\mathrm{g} \sum_{t=0}^{T-1} \|\nabla_{\boldsymbol{g}_t} \mathcal{L}(\theta_t)\|_2^2 \leq \mathcal{L}(\theta_0) - \mathcal{L}_\mathrm{min} + \sum_{i=1}^m K_i, \quad (20)$$

*where $K_i = \left( \frac{C_i \eta_\mathrm{w}}{\|\boldsymbol{w}_0^{(i)}\|_2^2} + \frac{1}{2} \right) \left( 2C_i + (\pi L_i^\mathrm{vv})^2 \frac{\eta_\mathrm{w}}{\|\boldsymbol{w}_0^{(i)}\|_2^2} \right) = \tilde{O}(\eta_\mathrm{w}^2 + 1)$.*

*Proof.* A.2 For every $i = 1, \ldots, m$, let $S_t^{(i)} := -\eta_\mathrm{w} \sum_{\tau=0}^{t-1} \|\nabla_{\boldsymbol{w}_\tau^{(i)}} \mathcal{L}(\theta_\tau)\|_2^2 \left( 1 - \frac{C_i \eta_\mathrm{w}}{\|\boldsymbol{w}_\tau^{(i)}\|_2^2} \right)$. Also let $G_t := -(1/2) c_\mathrm{g} \eta_\mathrm{g} \sum_{\tau=0}^{t-1} \|\nabla_{\boldsymbol{g}_t} \mathcal{L}(\theta_t)\|_2^2$. By Lemma 3.2 we know that $\mathcal{L}(\theta_T) \leq \mathcal{L}(\theta_0) + \sum_{i=1}^m S_T^{(i)} + G_T$.

**Upper Bound for Weight Scales at $t_i + 1$.** If $t_i = -1$, then $\|\boldsymbol{w}_{t_i+1}^{(i)}\|_2^2 = \|\boldsymbol{w}_0^{(i)}\|_2^2$. For $t_i \geq 0$, we can bound $\|\boldsymbol{w}_{t_i+1}^{(i)}\|_2^2$ by Lemma A.1:

$$\|\boldsymbol{w}_{t_i+1}^{(i)}\|_2^2 = \|\boldsymbol{w}_{t_i}^{(i)}\|_2^2 + \eta_\mathrm{w}^2 \left\| \nabla_{\boldsymbol{w}_{t_i}^{(i)}} \mathcal{L}(\theta_{t_i}) \right\|_2^2 \leq 2C_i \eta_\mathrm{w} + \eta_\mathrm{w}^2 \left( \frac{\pi L_i^\mathrm{vv}}{\|\boldsymbol{w}_0^{(i)}\|_2} \right)^2$$

$$= \eta_\mathrm{w} \left( 2C_i + (\pi L_i^\mathrm{vv})^2 \frac{\eta_\mathrm{w}}{\|\boldsymbol{w}_0^{(i)}\|_2^2} \right).$$

In either case, we have $\|\boldsymbol{w}_{t_i+1}^{(i)}\|_2^2 \leq \|\boldsymbol{w}_0^{(i)}\|_2^2 + \eta_\mathrm{w} \left( 2C_i + (\pi L_i^\mathrm{vv})^2 \frac{\eta_\mathrm{w}}{\|\boldsymbol{w}_0^{(i)}\|_2^2} \right)$.

**Upper Bound for Weight Scales at $T$.** Since $\|\boldsymbol{w}_\tau^{(i)}\|_2^2$ is non-decreasing with $\tau$ and $\|\boldsymbol{w}_{t_i+1}^{(i)}\|_2^2 > 2C_i \eta_\mathrm{w}$,

$$S_T^{(i)} - S_{t_i+1}^{(i)} = -\eta_\mathrm{w} \sum_{\tau=t_i'+1}^{T-1} \|\nabla_{\boldsymbol{w}_\tau^{(i)}} \mathcal{L}(\theta_\tau)\|_2^2 \left( 1 - \frac{C_i \eta_\mathrm{w}}{\|\boldsymbol{w}_\tau^{(i)}\|_2^2} \right)$$

$$\leq -\frac{\eta_\mathrm{w}}{2} \sum_{\tau=t_i'+1}^{T-1} \|\nabla_{\boldsymbol{w}_\tau^{(i)}} \mathcal{L}(\theta_\tau)\|_2^2$$

$$= -\frac{1}{2\eta_\mathrm{w}} \left( \|\boldsymbol{w}_T^{(i)}\|_2^2 - \|\boldsymbol{w}_{t_i'+1}^{(i)}\|_2^2 \right)$$

$$\leq -\frac{1}{2\eta_\mathrm{w}} \|\boldsymbol{w}_T^{(i)}\|_2^2 + \frac{1}{2\eta_\mathrm{w}} \left( \|\boldsymbol{w}_0^{(i)}\|_2^2 + \eta_\mathrm{w} \left( 2C_i + (\pi L_i^\mathrm{vv})^2 \frac{\eta_\mathrm{w}}{\|\boldsymbol{w}_0^{(i)}\|_2^2} \right) \right)$$

$$\leq -\frac{\|\boldsymbol{w}_T^{(i)}\|_2^2}{2\eta_\mathrm{w}} + \frac{\|\boldsymbol{w}_0^{(i)}\|_2^2}{2\eta_\mathrm{w}} + \frac{1}{2} \left( 2C_i + (\pi L_i^\mathrm{vv})^2 \frac{\eta_\mathrm{w}}{\|\boldsymbol{w}_0^{(i)}\|_2^2} \right),$$

where we use the fact that $\|\boldsymbol{w}_t^{(i)}\|_2^2 = \|\boldsymbol{w}_0^{(i)}\|_2^2 + \eta_{\mathrm{w}}^2 \sum_{\tau=0}^{t-1} \|\nabla_{\boldsymbol{w}_\tau^{(i)}} \mathcal{L}(\theta_\tau)\|_2^2$ at the third line. We can bound $S_{t_i+1}^{(i)}$ by

$$S_{t_i+1}^{(i)} \leq \eta_{\mathrm{w}} \left( \sum_{\tau=0}^{t_i} \|\nabla_{\boldsymbol{w}_\tau^{(i)}} \mathcal{L}(\theta_\tau)\|_2^2 \right) \frac{C_i \eta_{\mathrm{w}}}{\|\boldsymbol{w}_0^{(i)}\|_2^2} \leq (\|\boldsymbol{w}_{t_i+1}^{(i)}\|_2^2 - \|\boldsymbol{w}_0^{(i)}\|_2^2) \cdot \frac{C_i}{\|\boldsymbol{w}_0^{(i)}\|_2^2}$$

$$\leq \eta_{\mathrm{w}} \left( 2C_i + (\pi L_i^{\mathrm{vv}})^2 \frac{\eta_{\mathrm{w}}}{\|\boldsymbol{w}_0^{(i)}\|_2^2} \right) \cdot \frac{C_i}{\|\boldsymbol{w}_0^{(i)}\|_2^2}.$$

Combining them together, we have

$$\frac{\|\boldsymbol{w}_T^{(i)}\|_2^2 - \|\boldsymbol{w}_0^{(i)}\|_2^2}{2\eta_{\mathrm{w}}} \leq -S_T^{(i)} + S_{t_i+1}^{(i)} + \frac{1}{2} \left( 2C_i + (\pi L_i^{\mathrm{vv}})^2 \frac{\eta_{\mathrm{w}}}{\|\boldsymbol{w}_0^{(i)}\|_2^2} \right)$$

$$\leq -S_T^{(i)} + \left( \frac{C_i \eta_{\mathrm{w}}}{\|\boldsymbol{w}_0^{(i)}\|_2^2} + \frac{1}{2} \right) \left( 2C_i + (\pi L_i^{\mathrm{vv}})^2 \frac{\eta_{\mathrm{w}}}{\|\boldsymbol{w}_0^{(i)}\|_2^2} \right)$$

$$\leq -S_T^{(i)} + K_i.$$

Taking sum over all $i = 1, \ldots, m$ and also subtracting $G_T$ on the both sides, we have

$$\sum_{i=1}^m \frac{\|\boldsymbol{w}_T^{(i)}\|_2^2 - \|\boldsymbol{w}_0^{(i)}\|_2^2}{2\eta_{\mathrm{w}}} + (1/2)c_{\mathrm{g}}\eta_{\mathrm{g}} \sum_{t=0}^{T-1} \|\nabla_{\boldsymbol{g}_t} \mathcal{L}(\theta_t)\|_2^2 \leq -\sum_{i=1}^m S_T^{(i)} - G_T + \sum_{i=1}^m K_i$$

$$\leq \mathcal{L}(\theta_0) - \mathcal{L}_{\min} + \sum_{i=1}^m K_i.$$

where $\mathcal{L}_{\min} \leq \mathcal{L}(\theta_T) \leq \mathcal{L}(\theta_0) + \sum_{i=1}^m S_T^{(i)} + G_T$ is used at the second line. $\qquad \square$

Combining the lemmas above together, we can obtain our results.

*Proof for Theorem 3.1.* By Lemma A.2, we have

$$\sum_{i=1}^m \sum_{t=0}^{T-1} \left\| \nabla_{\boldsymbol{v}_t^{(i)}} \mathcal{L}(V_t; \boldsymbol{g}_t) \right\|_2^2 = \sum_{i=1}^m \sum_{t=0}^{T-1} \|\boldsymbol{w}_t^{(i)}\|_2^2 \left\| \nabla_{\boldsymbol{w}_t^{(i)}} \mathcal{L}(W_t; \boldsymbol{g}_t) \right\|_2^2$$

$$\leq \sum_{i=1}^m \|\boldsymbol{w}_T^{(i)}\|_2^2 \cdot \frac{\|\boldsymbol{w}_T^{(i)}\|_2^2 - \|\boldsymbol{w}_0^{(i)}\|_2^2}{\eta_{\mathrm{w}}^2}$$

$$\leq \max_{1 \leq i \leq m} \left\{ \frac{\|\boldsymbol{w}_T^{(i)}\|_2^2}{\eta_{\mathrm{w}}} \right\} \cdot \sum_{i=1}^m \frac{\|\boldsymbol{w}_T^{(i)}\|_2^2 - \|\boldsymbol{w}_0^{(i)}\|_2^2}{\eta_{\mathrm{w}}}$$

$$\leq 4 \left( \mathcal{L}(\theta_0) - \mathcal{L}_{\min} + \sum_{i=1}^m K_i + \sum_{i=1}^m \frac{\|\boldsymbol{w}_0^{(i)}\|_2^2}{\eta_{\mathrm{w}}} \right) \left( \mathcal{L}(\theta_0) - \mathcal{L}_{\min} + \sum_{i=1}^m K_i \right)$$

$$\leq \tilde{O} \left( \frac{1}{\eta_{\mathrm{w}}} + \eta_{\mathrm{w}}^4 \right).$$

$$\sum_{t=0}^{T-1} \|\nabla_{\boldsymbol{g}_t} \mathcal{L}(V_t; \boldsymbol{g}_t)\|_2^2 \leq \frac{2}{c_{\mathrm{g}}\eta_{\mathrm{g}}} \left( \mathcal{L}(\theta_0) - \mathcal{L}_{\min} + \sum_{i=1}^m K_i \right)$$

$$\leq \tilde{O} \left( \frac{1 + 1\eta_{\mathrm{w}}^2}{\eta_{\mathrm{g}}} \right).$$

Combining them together we have

$$
T \cdot \min_{0 \leq t < T} \left\{ \sum_{i=1}^{m} \left\| \nabla_{\boldsymbol{v}_t^{(i)}} \mathcal{L}(V_t; \boldsymbol{g}_t) \right\|_2^2 + \| \boldsymbol{g}_t \|_2^2 \right\}
$$

$$
\leq \sum_{i=1}^{m} \sum_{t=0}^{T-1} \left\| \nabla_{\boldsymbol{v}_t^{(i)}} \mathcal{L}(V_t; \boldsymbol{g}_t) \right\|_2^2 + \sum_{t=0}^{T-1} \| \nabla_{\boldsymbol{g}_t} \mathcal{L}(V_t; \boldsymbol{g}_t) \|_2^2
$$

$$
\leq \tilde{O} \left( \frac{1}{\eta_{\mathrm{w}}} + \eta_{\mathrm{w}}^4 + \frac{1 + \eta_{\mathrm{w}}^2}{\eta_{\mathrm{g}}} \right).
$$

Thus $\min_{0 \leq t < T} \| \nabla \mathcal{L}(V_t, g_t) \|_2$ converges in the rate of

$$
\min_{0 \leq t < T} \| \nabla \mathcal{L}(V_t, g_t) \|_2 = \tilde{O} \left( \frac{1}{\sqrt{T}} \left( \frac{1}{\sqrt{\eta_{\mathrm{w}}}} + \eta_{\mathrm{w}}^2 + \frac{1 + \eta_{\mathrm{w}}}{\sqrt{\eta_{\mathrm{g}}}} \right) \right).
$$

$\square$

## B    PROOF FOR STOCHASTIC GRADIENT DESCENT

Let $\mathscr{F}_t = \sigma\{\boldsymbol{z}_0, \ldots, \boldsymbol{z}_{t-1}\}$ be the filtration, where $\sigma\{\cdot\}$ denotes the sigma field.

We use $\mathcal{L}_t := \mathcal{L}_{\boldsymbol{z}_t}(\boldsymbol{\theta}_t), \mathcal{F}_t := \mathcal{F}_{\boldsymbol{z}_t}(\boldsymbol{\theta}_t)$ for simplicity. As usual, we define $\boldsymbol{v}_t^{(i)} = \boldsymbol{w}_t^{(i)}/\|\boldsymbol{w}_t^{(i)}\|_2$. We use the notations $\nabla_{\boldsymbol{v}_t^{(i)}} \mathcal{L}_t := \nabla_{\boldsymbol{v}_t^{(i)}} \mathcal{L}_{\boldsymbol{z}_t}(V_t, \boldsymbol{g}_t)$ and $\nabla_{\boldsymbol{v}_t^{(i)}} \mathcal{F}_t := \nabla_{\boldsymbol{v}_t^{(i)}} \mathcal{F}_{\boldsymbol{z}_t}(V_t, \boldsymbol{g}_t)$ for short. Let $\Delta \boldsymbol{w}_t^{(i)} = -\eta_{\mathrm{w},t} \nabla_{\boldsymbol{w}_t^{(i)}} \mathcal{F}_t, \Delta \boldsymbol{g}_t = -\eta_{\mathrm{g},t} \nabla_{\boldsymbol{g}_t} \mathcal{F}_t$.

**Lemma B.1.** *For any* $a_0, \ldots, a_T \in [0, B]$ *with* $a_0 > 0$,

$$
\sum_{t=1}^{T} \frac{a_t}{\sum_{\tau=0}^{t-1} a_\tau} \leq \log_2 \left( \sum_{t=0}^{T-1} a_t/a_0 \right) + 1 + 2B/a_0
$$

*Proof.* Let $t_i$ be the minimum $1 \leq t \leq T$ such that $\sum_{\tau=0}^{t-1} a_\tau \geq a_0 \cdot 2^i$. Let $k$ be the maximum $i$ such that $t_i$ exists. Let $t_{k+1} = T + 1$. Then we know that

$$
\sum_{t=t_i}^{t_{i+1}-1} \frac{a_t}{\sum_{\tau=0}^{t-1} a_\tau} \leq \frac{a_0 \cdot 2^i + B}{a_0 \cdot 2^i} \leq 1 + \frac{B}{a_0} \cdot 2^{-i}.
$$

Thus, $\sum_{t=1}^{T} \frac{a_t}{\sum_{\tau=0}^{t-1} a_\tau} \leq \sum_{i=0}^{k} \left( 1 + \frac{B}{a_0} \cdot 2^{-i} \right) = k + 1 + \frac{2B}{a_0} \leq \log_2 \left( \sum_{t=0}^{T-1} a_t/a_0 \right) + 1 + 2B/a_0$.

$\square$

**Lemma B.2.** *Fix* $T > 0$. *Let*

$$
C_i := \frac{1}{2} \sum_{j=1}^{m} L_{ij}^{\mathrm{vv}} + L_i^{\mathrm{vg}\,2} m/(c_{\mathrm{g}} L^{\mathrm{gg}})
$$

$$
U := \frac{1 + c_{\mathrm{g}}/2}{2} L^{\mathrm{gg}} G_{\mathrm{g}}^2
$$

$$
S_i := -\sum_{t=0}^{T-1} \frac{\eta_{\mathrm{w},t}}{\|\boldsymbol{w}_t^{(i)}\|_2^2} \| \nabla_{\boldsymbol{v}_t^{(i)}} \mathcal{L}_t \|_2^2 + C_i \sum_{t=0}^{T-1} \frac{\eta_{\mathrm{w},t}^2}{\|\boldsymbol{w}_t^{(i)}\|_2^4} \| \nabla_{\boldsymbol{v}_t^{(i)}} \mathcal{F}_t \|_2^2
$$

$$
R := -\frac{1}{2} c_{\mathrm{g}} \sum_{t=0}^{T-1} \eta_{\mathrm{g},t} \| \nabla_{\boldsymbol{g}_t} \mathcal{L}_t \|_2^2 + U \sum_{t=0}^{T-1} \eta_{\mathrm{g},t}^2 G_{\mathrm{g}}
$$

*Then* $\mathbb{E}[\mathcal{L}_T] - \mathcal{L}_0 \leq \sum_{i=1}^{m} \mathbb{E}[S_i] + \mathbb{E}[R]$.

*Proof.* Conditioned on $\mathscr{F}_t$, by Taylor expansion, we have

$$\mathbb{E}[\mathcal{L}_{t+1} \mid \mathscr{F}_t] \leq \mathcal{L}_t - \eta_{\mathrm{w},t} \sum_{i=1}^{m} \|\nabla_{\boldsymbol{w}_t^{(i)}} \mathcal{L}_t\|_2^2 - \eta_{\mathrm{g},t} \|\nabla_{\boldsymbol{g}_t} \mathcal{L}_t\|_2^2 + \mathbb{E}[Q_t \mid \mathscr{F}_t] \qquad (21)$$

where $Q_t$ is

$$Q_t = \frac{1}{2} \sum_{i=1}^{m} \sum_{j=1}^{m} \frac{L_{ij}^{\mathrm{vv}}}{\left\|\boldsymbol{w}_t^{(i)}\right\|_2 \left\|\boldsymbol{w}_t^{(j)}\right\|_2} \|\Delta \boldsymbol{w}_t^{(i)}\|_2 \|\Delta \boldsymbol{w}_t^{(j)}\|_2$$
$$+ \sum_{i=1}^{m} \frac{L_i^{\mathrm{vg}}}{\left\|\boldsymbol{w}_t^{(i)}\right\|_2} \|\Delta \boldsymbol{w}_t^{(i)}\|_2 \|\Delta \boldsymbol{g}_t\|_2 + \frac{1}{2} L^{\mathrm{gg}} \|\Delta \boldsymbol{g}_t\|_2^2$$

By the inequality $\sqrt{ab} \leq \frac{1}{2}a + \frac{1}{2}b$, we have

$$\|\Delta \boldsymbol{w}_t^{(i)}\|_2 \|\Delta \boldsymbol{w}_t^{(j)}\|_2 \frac{L_{ij}^{\mathrm{vv}}}{\|\boldsymbol{w}_t^{(i)}\|_2 \|\boldsymbol{w}_t^{(j)}\|_2} \leq \frac{1}{2} \|\Delta \boldsymbol{w}_t^{(i)}\|_2^2 \frac{L_{ij}^{\mathrm{vv}}}{\|\boldsymbol{w}_t^{(i)}\|_2^2} + \frac{1}{2} \|\Delta \boldsymbol{w}_t^{(j)}\|_2^2 \frac{L_{ij}^{\mathrm{vv}}}{\|\boldsymbol{w}_t^{(j)}\|_2^2}$$
$$\|\Delta \boldsymbol{w}_t^{(i)}\|_2 \|\Delta \boldsymbol{g}_t\|_2 \frac{L_i^{\mathrm{vg}}}{\|\boldsymbol{w}_t^{(i)}\|_2} \leq \|\Delta \boldsymbol{w}_t^{(i)}\|_2^2 \frac{L_i^{\mathrm{vg}\,2} m/(c_{\mathrm{g}} L^{\mathrm{gg}})}{\|\boldsymbol{w}_t^{(i)}\|_2^2} + \frac{1}{4} c_{\mathrm{g}} \|\Delta \boldsymbol{g}_t\|_2^2 \frac{L^{\mathrm{gg}}}{m}.$$

Note that $\mathbb{E}[\|\Delta \boldsymbol{g}_t\|_2^2 \mid \mathscr{F}_t] \leq \eta_{\mathrm{g},t}^2 \left(\|\nabla_{\boldsymbol{g}_t} \mathcal{F}_t\|_2^2 + G_{\mathrm{g}}^2\right)$. Thus,

$$\mathbb{E}[Q_t \mid \mathscr{F}_t] \leq \sum_{i=1}^{m} \frac{1}{2} \left(\sum_{j=1}^{m} L_{ij}^{\mathrm{vv}} + 2 L_i^{\mathrm{vg}\,2} m/(c_{\mathrm{g}} L^{\mathrm{gg}})\right) \frac{\eta_{\mathrm{w},t}^2}{\|\boldsymbol{w}_t^{(i)}\|_2^2} \|\nabla_{\boldsymbol{w}_t^{(i)}} \mathcal{F}_t\|_2^2$$
$$+ \frac{1 + c_{\mathrm{g}}/2}{2} L^{\mathrm{gg}} \cdot \eta_{\mathrm{g},t}^2 \left(\|\nabla_{\boldsymbol{g}_t} \mathcal{F}_t\|_2^2 + G_{\mathrm{g}}^2\right)$$
$$\leq \sum_{i=1}^{m} C_i \frac{\eta_{\mathrm{w},t}^2}{\|\boldsymbol{w}_t^{(i)}\|_2^2} \|\nabla_{\boldsymbol{w}_t^{(i)}} \mathcal{F}_t\|_2^2 + (1 - c_{\mathrm{g}}/2) \eta_{\mathrm{g},t} \|\nabla_{\boldsymbol{g}_t} \mathcal{F}_t\|_2^2 + U \eta_{\mathrm{g},t}^2$$

Taking this into equation 21 and summing up for all $t$, we have

$$\mathbb{E}[\mathcal{L}_T] - \mathcal{L}_0 \leq - \sum_{i=1}^{m} \sum_{t=0}^{T-1} \mathbb{E}\left[\frac{\eta_{\mathrm{w},t}}{\|\boldsymbol{w}_t^{(i)}\|_2^2} \|\nabla_{\boldsymbol{v}_t^{(i)}} \mathcal{L}_t\|_2^2\right] - \frac{1}{2} c_{\mathrm{g}} \sum_{t=0}^{T-1} \mathbb{E}\left[\eta_{\mathrm{g},t} \|\nabla_{\boldsymbol{g}_t} \mathcal{L}_t\|_2^2\right]$$
$$+ \sum_{i=1}^{m} C_i \sum_{t=0}^{T-1} \mathbb{E}\left[\frac{\eta_{\mathrm{w},t}^2}{\|\boldsymbol{w}_t^{(i)}\|_2^2} \|\nabla_{\boldsymbol{w}_t^{(i)}} \mathcal{F}_t\|_2^2\right] + U \sum_{t=0}^{T-1} \eta_{\mathrm{g},t}^2 G_{\mathrm{g}},$$

and the right hand side can be expressed as $\sum_{i=1}^{m} \mathbb{E}[S_i] + \mathbb{E}[R]$ by definitions. $\qquad \square$

**Lemma B.3.** *For any $T \geq 0$, $1 \leq i \leq m$, we have*

$$\frac{\eta_{\mathrm{w},T}}{\|\boldsymbol{w}_T^{(i)}\|_2^2} \geq \begin{cases} \tilde{\Omega}(T^{-1/2}) & \text{if } 0 \leq \alpha < 1/2; \\ \tilde{\Omega}((T \log T)^{-1/2}) & \text{if } \alpha = 1/2. \end{cases}$$

*Proof.* By Lemma 2.4, we have

$$\|\boldsymbol{w}_{t+1}^{(i)}\|_2^4 = \left(\|\boldsymbol{w}_t^{(i)}\|_2^2 + \frac{\eta_{\mathrm{w},t}^2}{\|\boldsymbol{w}_t^{(i)}\|_2^2}\|\nabla_{\boldsymbol{v}_t^{(i)}}\mathcal{F}_t\|_2^2\right)^2$$

$$= \|\boldsymbol{w}_t^{(i)}\|_2^4 + 2\eta_{\mathrm{w},t}^2\|\nabla_{\boldsymbol{v}_t^{(i)}}\mathcal{F}_t\|_2^2 + \frac{\eta_{\mathrm{w},t}^4}{\|\boldsymbol{w}_t^{(i)}\|_2^4}\|\nabla_{\boldsymbol{v}_t^{(i)}}\mathcal{F}_t\|_2^4$$

$$\leq \|\boldsymbol{w}_t^{(i)}\|_2^4 + 2\eta_{\mathrm{w}}^2(t+1)^{-2\alpha}\left(\pi L_{ii}^{\mathrm{vv}}\right)^2 + \eta_{\mathrm{w}}^4(t+1)^{-4\alpha}\left(\frac{\pi L_{ii}^{\mathrm{vv}}}{\|\boldsymbol{w}_0^{(i)}\|_2}\right)^4$$

$$\leq \|\boldsymbol{w}_t^{(i)}\|_2^4 + (t+1)^{-2\alpha}\left(2\eta_{\mathrm{w}}^2\left(\pi L_{ii}^{\mathrm{vv}}\right)^2 + \eta_{\mathrm{w}}^4\left(\frac{\pi L_{ii}^{\mathrm{vv}}}{\|\boldsymbol{w}_0^{(i)}\|_2}\right)^4\right)$$

$$\leq \|\boldsymbol{w}_0^{(i)}\|_2^4 + \left(\sum_{\tau=0}^{t}\frac{1}{(\tau+1)^{2\alpha}}\right)\left(2\eta_{\mathrm{w}}^2\left(\pi L_{ii}^{\mathrm{vv}}\right)^2 + \eta_{\mathrm{w}}^4\left(\frac{\pi L_{ii}^{\mathrm{vv}}}{\|\boldsymbol{w}_0^{(i)}\|_2}\right)^4\right).$$

For $0 \leq \alpha < 1/2$, $\sum_{\tau=0}^{T-1}\frac{1}{(\tau+1)^{2\alpha}} = O(T^{1-2\alpha})$, so $\frac{\eta_{\mathrm{w},T}}{\|\boldsymbol{w}_T^{(i)}\|_2^2} = \tilde{\Omega}(T^{-1/2})$; for $\alpha = 1/2$, $\sum_{\tau=0}^{T-1}\frac{1}{(\tau+1)^{2\alpha}} = O(\log T)$, so $\frac{\eta_{\mathrm{w},T}}{\|\boldsymbol{w}_T^{(i)}\|_2^2} = \tilde{\Omega}((T\log T)^{-1/2})$. □

**Lemma B.4.** *For $T > 0$,*

- *$S_i$ can be bounded by $S_i \leq -\frac{\eta_{\mathrm{w},T}}{\|\boldsymbol{w}_T^{(i)}\|_2^2}\sum_{t=0}^{T-1}\|\nabla_{\boldsymbol{v}_t^{(i)}}\mathcal{L}_t\|_2^2 + \tilde{O}(\log T)$;*

- *$R$ can be bounded by $R \leq -\tilde{\Omega}(T^{-1/2})\sum_{t=0}^{T-1}\|\nabla_{\boldsymbol{g}_t}\mathcal{L}_t\|_2^2 + \tilde{O}(\log T)$.*

*Proof.* Fix $i \in \{1,\ldots,m\}$. First we bound $S_i$. Recall that

$$S_i := -\sum_{t=0}^{T-1}\frac{\eta_{\mathrm{w},t}}{\|\boldsymbol{w}_t^{(i)}\|_2^2}\|\nabla_{\boldsymbol{v}_t^{(i)}}\mathcal{L}_t\|_2^2 + C_i\sum_{t=0}^{T-1}\frac{\eta_{\mathrm{w},t}^2}{\|\boldsymbol{v}_t^{(i)}\|_2^2}\|\nabla_{\boldsymbol{v}_t^{(i)}}\mathcal{F}_t\|_2^2.$$

Note that $\frac{\eta_{\mathrm{w},t}}{\|\boldsymbol{w}_t^{(i)}\|_2^2}$ is non-increasing, so

$$-\sum_{t=0}^{T-1}\frac{\eta_{\mathrm{w},t}}{\|\boldsymbol{w}_t^{(i)}\|_2^2}\|\nabla_{\boldsymbol{v}_t^{(i)}}\mathcal{L}_t\|_2^2 \leq -\frac{\eta_{\mathrm{w},T}}{\|\boldsymbol{w}_T^{(i)}\|_2^2}\sum_{t=0}^{T-1}\|\nabla_{\boldsymbol{v}_t^{(i)}}\mathcal{L}_t\|_2^2. \tag{22}$$

Also note that $\|\boldsymbol{w}_t^{(i)}\|_2^2 \geq \|\boldsymbol{w}_0^{(i)}\|_2^2 + \sum_{\tau=0}^{t-1}\eta_{\mathrm{w},\tau}^2\|\nabla_{\boldsymbol{w}_t^{(i)}}\mathcal{F}_t\|_2^2$. By Lemma B.1,

$$C_i\sum_{t=0}^{T-1}\frac{\eta_{\mathrm{w},t}^2}{\|\boldsymbol{w}_t^{(i)}\|_2^2}\|\nabla_{\boldsymbol{w}_t^{(i)}}\mathcal{F}_t\|_2^2 \leq C_i\sum_{t=0}^{T-1}\frac{\eta_{\mathrm{w},t}^2\|\nabla_{\boldsymbol{w}_t^{(i)}}\mathcal{F}_t\|_2^2}{\|\boldsymbol{w}_0^{(i)}\|_2^2 + \sum_{\tau=0}^{t-1}\eta_{\mathrm{w},\tau}^2\|\nabla_{\boldsymbol{w}_t^{(i)}}\mathcal{F}_t\|_2^2}$$

$$\leq C_i\left(\log_2\left(\frac{\|\boldsymbol{w}_T^{(i)}\|_2^2}{\|\boldsymbol{w}_0^{(i)}\|_2^2}\right) + 1 + \frac{2\eta_{\mathrm{w}}^2(\pi L_{ii}^{\mathrm{vv}})^2}{\|\boldsymbol{w}_0^{(i)}\|_2^4}\right) = \tilde{O}(\log T). \tag{23}$$

We can get the bound for $S_i$ by combining equation 22 and equation 23.

Now we bound $R$. Recall that

$$R := -\frac{1}{2}c_{\mathrm{g}}\sum_{t=0}^{T-1}\eta_{\mathrm{g},t}\|\nabla_{\boldsymbol{g}_t}\mathcal{L}_t\|_2^2 + U\sum_{t=0}^{T-1}\eta_{\mathrm{g},t}^2 G_{\mathrm{g}}.$$

The first term can be bounded by $-\tilde{\Omega}(T^{-1/2})\sum_{t=0}^{T-1}\|\nabla_{\boldsymbol{g}_t}\mathcal{L}_t\|_2^2$ by noticing that $\eta_{\mathrm{g},t} \geq \eta_{\mathrm{g},T} = \Omega(T^{-1/2})$. The second term can be bounded by $\tilde{O}(\log T)$ since $\sum_{t=0}^{T-1}\eta_{\mathrm{g},t}^2 = \sum_{t=0}^{T-1}\tilde{O}(1/t) = \tilde{O}(\log T)$. □

*Proof for Theorem 4.2.* Combining Lemma B.2 and Lemma B.4, for $0 \leq \alpha < 1/2$, we have

$$\mathcal{L}_{\min} - \mathcal{L}_0 \leq \mathbb{E}[\mathcal{L}_T] - \mathcal{L}_0 \leq -\tilde{\Omega}(T^{-1/2}) \sum_{t=0}^{T-1} \mathbb{E}\left[\|\nabla\mathcal{L}(V_t; \boldsymbol{g}_t)\|_2^2\right] + \tilde{O}(\log T).$$

Thus,

$$\min_{0 \leq t < T} \mathbb{E}\left[\|\nabla\mathcal{L}(V_t; \boldsymbol{g}_t)\|_2^2\right] \leq \frac{1}{T}\sum_{t=0}^{T-1} \mathbb{E}\left[\|\nabla\mathcal{L}(V_t; \boldsymbol{g}_t)\|_2^2\right] \leq \tilde{O}\left(\frac{\log T}{\sqrt{T}}\right).$$

Similarly, for $\alpha = 1/2$, we have

$$\mathcal{L}_{\min} - \mathcal{L}_0 \leq \mathbb{E}[\mathcal{L}_T] - \mathcal{L}_0 \leq -\tilde{\Omega}((T\log T)^{-1/2}) \sum_{t=0}^{T-1} \mathbb{E}\left[\|\nabla\mathcal{L}(V_t; \boldsymbol{g}_t)\|_2^2\right] + \tilde{O}(\log T).$$

Thus,

$$\min_{0 \leq t < T} \mathbb{E}\left[\|\nabla\mathcal{L}(V_t; \boldsymbol{g}_t)\|_2^2\right] \leq \frac{1}{T}\sum_{t=0}^{T-1} \mathbb{E}\left[\|\nabla\mathcal{L}(V_t; \boldsymbol{g}_t)\|_2^2\right] \leq \tilde{O}\left(\frac{(\log T)^{3/2}}{\sqrt{T}}\right).$$

$\square$

## C  PROOF FOR THE SMOOTHNESS OF THE MOTIVATING NEURAL NETWORK

In this section we prove that the modified version of the motivating neural network does meet the assumptions in Section 2.4. More specifically, we assume:

- We use the network structure $\Phi$ in Section 2.1 with the smoothed variant of BN as described in Section 2.4;

- The objective $f_y(\cdot)$ is twice continuously differentiable, lower bounded by $f_{\min}$ and Lipschitz ($|f_y'(\hat{y})| \leq \alpha_f$);

- The activation $\sigma(\cdot)$ is twice continuously differentiable and Lipschitz ($|f_y'(\hat{y})| \leq \alpha_\sigma$);

- We add an extra weight decay (L2 regularization) term $\frac{\lambda}{2}\|\boldsymbol{g}\|_2^2$ to the loss in equation 2 for some $\lambda > 0$.

First, we show that $\boldsymbol{g}_t$ (containing all scale and shift parameters in BN) is bounded during the training process. Then the smoothness follows compactness using Extreme Value Theorem.

We use the following lemma to calculate back propagation:

**Lemma C.1.** *Let $\boldsymbol{x}_1, \ldots, \boldsymbol{x}_B$ be a set of vectors. Let $\boldsymbol{u} := \mathbb{E}_{b \in [B]}[\boldsymbol{x}_b]$ and $\boldsymbol{S} := \text{Var}_{b \in [B]}(\boldsymbol{x}_b)$. Let*

$$z_b := \gamma \frac{\boldsymbol{w}^\top(\boldsymbol{x}_b - \boldsymbol{u})}{\|\boldsymbol{w}\|_{S+\epsilon I}} + \beta.$$

*Let $y := f(z_1, \ldots, z_B) = f(\boldsymbol{z})$ for some function $f$. If $\|\nabla f(\boldsymbol{z})\|_2 \leq G$, then*

$$\left|\frac{\partial y}{\partial \gamma}\right| \leq G\sqrt{B} \qquad \left|\frac{\partial y}{\partial \beta}\right| \leq G\sqrt{B} \qquad \|\nabla_{\boldsymbol{x}_b} y\|_2 \leq \frac{3\gamma}{\epsilon} G\sqrt{B}.$$

*Proof.* Let $\tilde{\boldsymbol{x}} \in \mathbb{R}^B$ be the vector where $\tilde{x}_b := (\boldsymbol{w}^\top(\boldsymbol{x}_b - \boldsymbol{u}))/\|\boldsymbol{w}\|_{S+\epsilon I}$. It is easy to see $\|\tilde{\boldsymbol{x}}\|_2^2 \leq B$. Then

$$\left|\frac{\partial y}{\partial \gamma}\right| = \left|\nabla f(\boldsymbol{z})^\top \tilde{\boldsymbol{x}}\right| \leq G\|\tilde{\boldsymbol{x}}\|_2 \leq G\sqrt{B}.$$

$$\left|\frac{\partial y}{\partial \beta}\right| \leq \|\nabla f(\boldsymbol{z})\|_1 \leq G\sqrt{B}.$$

For $\boldsymbol{x}_b$, we have

$$\nabla_{\boldsymbol{x}_b}\boldsymbol{u} = \frac{1}{B}\boldsymbol{I}$$

$$\nabla_{\boldsymbol{x}_b}\|\boldsymbol{w}\|_{S+\epsilon\boldsymbol{I}}^2 = \nabla_{\boldsymbol{x}_b}\left(\frac{1}{B}\sum_{b'=1}^B (\boldsymbol{x}_{b'}^\top\boldsymbol{w})^2 - \left(\frac{1}{B}\sum_{b'=1}^B \boldsymbol{x}_{b'}^\top\boldsymbol{w}\right)^2\right)$$

$$= \frac{2}{B}\boldsymbol{w}\boldsymbol{w}^\top(\boldsymbol{x}_b - \boldsymbol{u}).$$

Then

$$\nabla_{\boldsymbol{x}_b}z_{b'} = \gamma\frac{(\boldsymbol{1}_{b=b'} - 1/B)\|\boldsymbol{w}\|_{S+\epsilon I}\boldsymbol{w} - \boldsymbol{w}^\top(\boldsymbol{x}_b - \boldsymbol{u})\cdot\frac{1}{\|\boldsymbol{w}\|_{S+\epsilon I}}\cdot\frac{2}{B}\boldsymbol{w}\boldsymbol{w}^\top(\boldsymbol{x}_b - \boldsymbol{u})}{\|\boldsymbol{w}\|_{S+\epsilon I}^2}$$

Since $(\boldsymbol{w}^\top(\boldsymbol{x}_b - \boldsymbol{u}))^2 \leq B\|\boldsymbol{w}\|_{S+\epsilon I}^2$,

$$\|\nabla_{\boldsymbol{x}_b}z_{b'}\|_2 \leq \gamma\left(\frac{1}{\|\boldsymbol{w}\|_{S+\epsilon I}}(\boldsymbol{1}_{b=b'} - 1/B)\|\boldsymbol{w}\|_2 + \frac{1}{\|\boldsymbol{w}\|_{S+\epsilon I}}\cdot 2\|\boldsymbol{w}\|_2\right) \leq \frac{3\gamma}{\epsilon}.$$

Thus,

$$\|\nabla_{\boldsymbol{x}_b}y\|_2 \leq \left(\sum_{b'=1}^B\left|\frac{\partial y}{\partial z_b}\right|^2\right)^{1/2}\left(\sum_{b'=1}^B\|\nabla_{\boldsymbol{x}_b}z_{b'}\|_2^2\right)^{1/2} \leq \frac{3\gamma}{\epsilon}G\sqrt{B}.$$

$\square$

**Lemma C.2.** *If $\|\boldsymbol{g}_0\|_2$ is bounded by a constant, there exists some constant $K$ such that $\|\boldsymbol{g}_t\|_2 \leq K$.*

*Proof.* Fix a time $t$ in the training process. Consider the process of back propagation. Define

$$R_i = \sum_{b=1}^B\sum_{k=1}^{m_i}\left(\frac{\partial}{\partial x_{b,k}^{(i)}}\mathcal{F}_{\boldsymbol{z}}(\theta)\right)^2,$$

where $x_{b,k}^{(i)}$ is the output of the $k$-th neuron in the $i$-th layer in the $b$-th data sample in the batch. By the Lipschitzness of the objective, $R_L$ can be bounded by a constant. If $R_i$ can be bounded by a constant, then by the Lipschitzness of $\sigma$ and Lemma C.1, the gradient of $\gamma$ and $\beta$ in layer $i$ can also be bounded by a constant. Note that

$$g_{t+1,k} = g_{t,k} - \eta_{\mathrm{g},t}\frac{\partial}{\partial g_{t,k}}\mathcal{F}_{\boldsymbol{z}}(\boldsymbol{\theta}_t) - \lambda\eta_{\mathrm{g},t}g_{t,k}$$

Thus $\gamma$ and $\beta$ in layer $i$ can be bounded by a constant since

$$|g_{t+1,k}| \leq (1 - \eta_{\mathrm{g},t}\lambda)|g_{t,k}| + \eta_{\mathrm{g},t}\lambda\cdot\frac{1}{\lambda}\left|\frac{\partial}{\partial g_{t,k}}\mathcal{F}_{\boldsymbol{z}}(\boldsymbol{\theta}_t)\right|.$$

Also Lemma C.1 and the Lipschitzness of $\sigma$ imply that $R_{i-1}$ can be bounded if $R_i$ and $\gamma$ in the layer $i$ can be bounded by a constant. Using a simple induction, we can prove the existence of $K$ for bounding the norm of $\|\boldsymbol{g}_t\|_2$ for all time $t$. $\square$

**Theorem C.3.** *If $\|\boldsymbol{g}_0\|_2$ is bounded by a constant, then $\Phi$ satisfies the assumptions in Section 2.4.*

*Proof.* Let $C$ be the set of parameters $\boldsymbol{\theta}$ satisfying $\|\boldsymbol{g}\| \leq K$ and $\|\boldsymbol{w}^{(i)}\|_2 = 1$ for all $1 \leq i \leq m$. By Lemma C.2, $C$ contains the set of $\tilde{\boldsymbol{\theta}}$ associated with the points lying between each pair of $\boldsymbol{\theta}_t$ and $\boldsymbol{\theta}_{t+1}$ (including the endpoints).

It is easy to show that $\mathcal{F}_{\boldsymbol{z}}(\tilde{\boldsymbol{\theta}})$ is twice continuously differentiable. Since $C$ is compact, by the Extreme Value Theorem, there must exist such constants $L_{ij}^{\mathrm{vv}}, L_i^{\mathrm{vg}}, L^{\mathrm{gg}}, G_{\mathrm{g}}$ for upper bounding the smoothness and the difference of gradients. $\square$

## D  EXPERIMENTS

In this section, we provide experimental evidence showing that the auto rate-tuning behavior does empower BN in the optimization aspect.

We trained a modified version of VGGNet (Simonyan & Zisserman, 2014) on Tensorflow. This network has $2 \times \mathrm{conv}64$, pooling, $3 \times \mathrm{conv}128$, pooling, $3 \times \mathrm{conv}256$, pooling, $3 \times \mathrm{conv}512$, pooling, $3 \times \mathrm{conv}512$, pooling, fc512, fc10 layers in order. Each convolutional layer has kernel size $3 \times 3$ and stride 1. ReLU is used as the activation function after each convolutional or fully-connected layer. We add a BN layer right before each ReLU. We set $\epsilon = 0$ in each BN, since we observed that the network works equally well for $\epsilon$ being 0 or an small number (such as $10^{-3}$, the default value in Tensorflow). We initialize the parameters according to the default configuration in Tensorflow: all the weights are initialized by Glorot uniform initializer (Glorot & Bengio, 2010); $\beta$ and $\gamma$ in BN are initialized by 0 and 1, respectively.

In this network, every kernel is scale-invariant, and for every BN layer except the last one, the concatenation of all $\beta$ and $\gamma$ parameters in this BN is also scale-invariant. Only $\beta$ and $\gamma$ parameters in the last BN are scale-variant (See Section 2.1). We consider the training in following two settings:

1. Train the network using the standard SGD (No momentum, learning rate decay, weight decay and dropout);

2. Train the network using Projected SGD (PSGD): at each iteration, one first takes a step proportional to the negative of the gradient calculated in a random batch, and then projects each scale-invariant parameter to the sphere with radius equal to its 2-norm before this iteration, i.e., rescales each scale-invariant parameter so that each maintains its length during training.

Note that the projection in Setting 2 removes the adaptivity of the learning rates in the corresponding intrinsic optimization problem, i.e., $G_t^{(i)}$ in equation 9 remains constant during the training. Thus, by comparing Setting 1 and Setting 2, we can know whether or not the auto-tuning behavior of BN shown in theory is effective in practice.

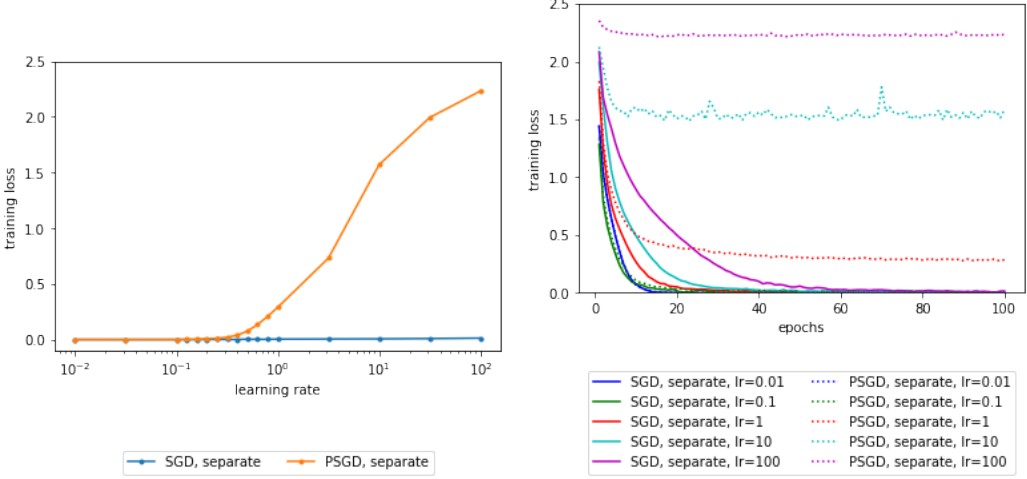

Figure 1: The relationship between the training loss and the learning rate for scale-invariant parameters, with learning rate for scale-variant ones set to 0.1. **Left:** The average training loss of the last 5 epochs (averaged across 10 experiments). In rare cases, the training loss becomes NaN in the experiments for the yellow curve (SGD, separate) with learning rate larger than 10. **Right:** The average training loss of each epoch (each curve stands for a single experiment).

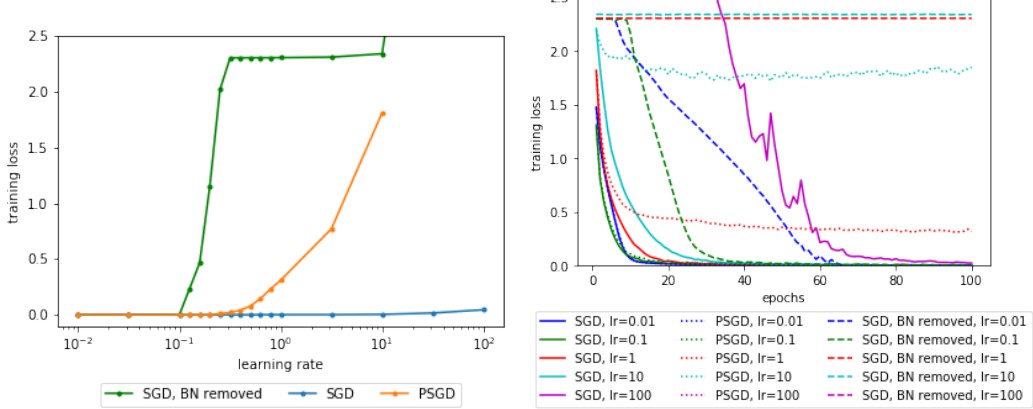

Figure 2: The relationship between the training loss and the learning rate. For learning rate larger than 10, the training loss of PSGD or SGD with BN removed is always either very large or NaN, and thus not invisible in the figure. **Left:** The average training loss of the last 5 epochs (averaged across 10 experiments). In rare cases, the training loss becomes NaN in the experiments for the green curve (`SGD, BN removed`) with learning rate larger than $10^{-0.7}$. We removed such data when taking the average. **Right:** The average training loss of each epoch (each curve stands for a single experiment).

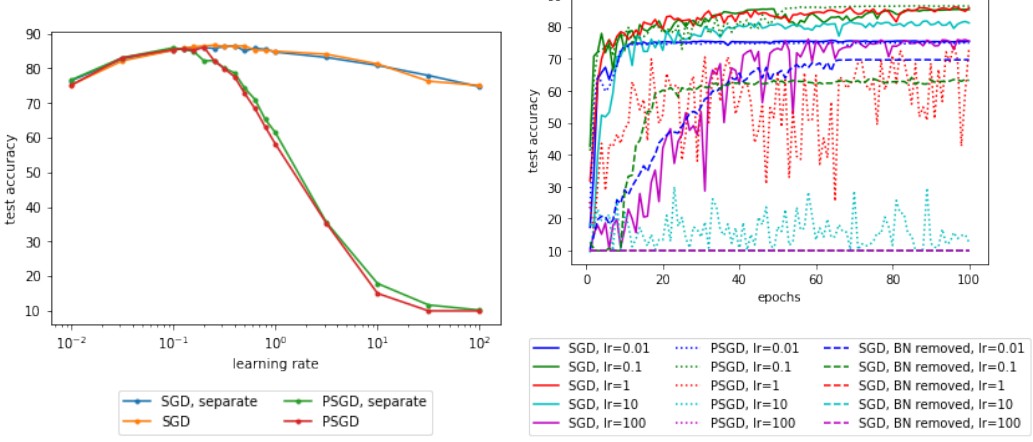

Figure 3: The relationship between the test accuracy and the learning rate. **Left:** The average test accuracy of the last 5 epochs (averaged across 10 experiments). **Right:** The test accuracy after each epoch (each curve stands for a single experiment). Due to the implementation of Tensorflow, outputting NaN leads to a test accuracy of $10\%$. Note that the magenta dotted curve (`PSGD, lr=100`), red dashed curve (`SGD, BN removed, lr=1`) and cyan dashed curve (`SGD, BN removed, lr=10`) are covered by the magenta dashed curve (`SGD, BN removed, lr=100`). They all have $10\%$ test accuracy.

## D.1 SEPARATE LEARNING RATES

As in our theoretical analysis, we consider what will happen if we set two learning rates separately for scale-invariant and scale-variant parameters. We train the network in either setting with different learning rates ranging from $10^{-2}$ to $10^2$ for 100 epochs.

First, we fix the learning rate for scale-variant ones to $0.1$, and try different learning rates for scale-invariant ones. As shown in Figure 1, for small learning rates (such as $0.1$), the training processes of networks in Setting 1 and 2 are very similar. But for larger learning rates, networks in Setting 1 can still converge to $0$ for all the learning rates we tried, while networks in Setting 2 got stuck with relatively large training loss. This suggests that the auto-tuning behavior of BN does takes effect when the learning rate is large, and it matches with the claimed effect of BN in (Ioffe & Szegedy, 2015) that BN enables us to use a higher learning rate. Though our theoretical analysis cannot be directly applied to the network we trained due to the non-smoothness of the loss function, the experiment results match with what we expect in our analysis.

## D.2 UNIFIED LEARNING RATE

Next, we consider the case in which we train the network with a unified learning rate for both scale-invariant and scale-variant parameters. We also compare Setting 1 and 2 with the setting in which we train the network with all the BN layers removed using SGD (we call it Setting 3).

As shown in Figure 2, the training loss of networks in Setting 1 converges to $0$. On the contrast, the training loss of networks in Setting 2 and 3 fails to converge to $0$ when a large learning rate is used, and in some cases the loss diverges to infinity or NaN. This suggests that the auto-tuning behavior of BN has an effective role in the case that a unified learning rate is set for all parameters.

For a fair comparison, we also trained neural networks in Setting 3 with initialization essentially equivalent to the ones with BN. This is done in the same way as (Krähenbühl et al., 2015; Mishkin & Matas, 2015) and Section 3 of (Salimans & Kingma, 2016): we first randomly initialize the parameters, then feed the first batch into the network and adjust the scaling and bias of each neuron to make its outputs have zero mean and unit variance. In this way, the loss of the networks converges to $0$ when the learning rate is smaller than $10^{-2.0}$, but for a slightly larger learning rate such as $10^{-1.8}$, the loss fails to converge to $0$, and sometimes even diverges to infinity or NaN. Compared with experimental results in Setting 1, this suggests that the robustness of training brought by BN is independent of the fact that BN changes the effective initialization of parameters.

## D.3 GENERALIZATION

Despite in Setting 1 the convergence of training loss for different learning rates, the convergence points can be different, which lead to different performances on test data.

In Figure 3, we plot the test accuracy of networks trained in Setting 1 and 2 using different unified learning rates, or separate learning rates with the learning rate for scale-variant parameters fixed to $0.1$. As shown in the Figure 3, the test accuracy of networks in Setting 2 decreases as the learning rate increases over $0.1$, while the test accuracy of networks in Setting 1 remains higher than $75\%$. The main reason that the network in Setting 2 doesn't perform well is underfitting, i.e. the network in Setting 2 fails to fit the training data well when learning rate is large. This suggests that the auto-tuning behavior of BN also benefits generalization since such behavior allows the algorithm to pick learning rates from a wider range while still converging to small test error.

