# OpenReview forum: "Theoretical Analysis of Auto Rate-Tuning by Batch Normalization"
_ICLR.cc/2019/Conference_

### Official Review · AnonReviewer1 · 2018-11-02
**A  theoretical result about asymptotic convergence with normalization but weakly related to the practical success of BN**

**Rating:** 7
**Confidence:** 4

**Review:**

* Description

The work is motivated by the empirical performance of Batch Normalization and in particular the observed better robustness of the choice of the learning rate.  Authors analyze theoretically the asymptotic convergence rate for objectives involving normalization, not necessarily BN, and show that for scale-invariant groups of parameters (appearing as a result of normalization) the initial learning rate may be set arbitrary while still asymptotic convergence is guaranteed with the same rate as the best known in the general case. Offline gradient descent and stochastic gradient descent cases are considered.

* Strengths

The work addresses better theoretical understanding of successful heuristics in deep learning, namely batch normalization and other normalizations. The technical results obtained are non-trivial and detailed proofs are presented. Also I did not verify the proofs the paper appears technically correct and technically clear. The result may be interpreted in the following form: if one chooses to use BN or other normalization, the paper gives a recommendation that only the learning rate of scale-variant parameters need to be set, which may have some practical advantages. Perhaps more important than the rate of convergence, is the guarantee that the method will not diverge (and will not get stuck in a non-local minimum).

* Criticism
This paper presents non-trivial theoretical results that are worth to be published but as I argue below its has a weak relevance to practice and the applicability of the obtained results is unclear.
-- Concerns regarding the clarity of presentation and interpretation of the results.

The properties of BN used as motivation for the study, are observed non-asymptotically with constant or empirically decreased learning rate schedules for a limited number of iterations. In contrast, the studied learning rates are asymptotic and there is a big discrepancy. SGD is observed to be significantly faster than batch gradient when far from convergence (experimental evidence), and this is with or without normalization. In practice, the training is stopped much before convergence, in the hope of finding solutions close to minimum with high probability. There is in fact no experimental evidence that the practical advantages of BN are relevant to the results proven. It makes a nice story that the theoretical properties justify the observations, but they may be as well completely unrelated.

As seen from the formal construction, the theoretical results apply equally well to all normalization methods. It occludes the clarity that BN is emphasized amongst them.

Considering theoretically, what advantages truly follow from the paper for optimizing a given function? Let’s consider the following cases.
1. For optimizing a general smooth function with all parameters forming a single scale-invariant vector. In this case, the paper proves that no careful selection of the learning rate is necessary. This result is beyond machine learning and unfortunately I cannot evaluate its merit. Is it known / not known in optimization?

2. The case of data-independent normalization (such as weight normalization).
Without normalization, we have to tune learning rate to achieve the optimal convergence. With normalization we still have to tune the learning rate (as scale-variant parameters remain or are reintroduced with each invariance to preserve the degrees of freedom), then we have to wait for the phase two of Lemma 3.2 so that the learning rate of scale-invariant parameters adapts, and from then on the optimal convergence rate can be guaranteed.

3. The case of Batch Normalization. Note that there is no direct correspondence between the loss of BN-normalized network (2) and the loss of the original network because of dependence of the normalization on the batches. In other words, there is no setting of parameters of the original network that would make its forward pass equivalent to that of BN network (2) for all batches. The theory tells the same as in case 2 above but with an additional price of optimizing a different function.

These points remain me puzzled regarding either practical or theoretical application of the result. It would be great if authors could elaborate.


-- Difference from Wu et al. 2018

This works is cited as a source of inspiration in several places in the paper. As the submission is a theoretical result with no immediate applicability, it would be very helpful if the authors could detail the technical improvements over this related work. Note, ICLR policy says that arxiv preprints earlier than one month before submission are considered a prior art. Could the authors elaborate more on possible practical/theoretical applications?


* Side Notes (not affecting the review recommendation)

I believe that the claim that “BN reduces covariate shift” (actively discussed in the intro) was an imprecise statement in the original work. Instead, BN should be able to quickly adapt to the covariate shift when it occurs. It achieves this by using the parameterization in which the mean and variance statistics of neurons (the quantities whose change is called the covariate shift) depend on variables that are local to the layer (gamma, beta in (1)) rather than on the cumulative effect of all of the preceding layers.

* Revision
I took into account the discussion and the newly added experiments and increased the score. The experiments verify the proven effect and make the paper more substantial. Some additional comments about experiments follow.
Training loss plots would be more clear in the log scale.
Comparison to "SGD BN removed" is not fair because the initialization is different (application of BN re-initializes weight scales and biases). The same initialization can be achieved by performing one training pass with BN with 0 learning rate and then removing it, see e.g. Gitman, I. and Ginsburg, B. (2017). Comparison of batch normalization and weight normalization algorithms for the large-scale image classification.
The use of Glorot uniform initializer is somewhat subtle. Since BN is used, Glorot initialization has no effect for a forward pass. However, it affects the gradient norm. Is there a rationale in this setting or it is just a more tricky method to fix the weight norm to some constant, e.g. ||w||=1?

---

> ### Author Response · Authors · 2018-11-18
> **Thanks for your careful review.**
>
> Thanks for your careful review! As mentioned in the intro, we are trying to give some principled insight into benefits of BN, which has proved tricky. Also, it is noted in the paper that BN probably has many desirable properties, of which auto-rate tuning is just one.
>
> (i) Speed of SGD vs GD:
> Note that “time” here refers to number of iterations, not epochs.  We are not aware of results establishing SGD is faster in this measure. (As noted on p2,  we are working within the standard paradigm of convergence rates in optimization. The only new part is the automatic rate tuning  behavior shown for most parameters when BN is used.)
>
> (ii) “usually training is stopped much before convergence, in the hope of finding solutions close to minimum with high probability.”
> We’re assuming training proceeds until gradient is small (stationary point). We are not aware of any prior analysis of speed of convergence that deviates from this assumption. Perhaps the reviewer is thinking of early stopping in context of better generalization?
>
> (iii) “clarify difference from Wu et al. (2018)”
> Wu et al. 2018 introduces a *new* algorithm inspired by weight normalization (WN) and studies its convergence rate to stationary point. This algorithm can be seen as an explicit way to tune the learning rate (thus it is conceptually analogous Adagrad). They don't have any results about WN or BN itself. Their analysis could be adapted to GD on one-neuron network with WN or BN without scale-variant parameters (gamma and beta). Even this adaptation is not immediate because the goal of this work is to find a stationary point on the unit sphere rather than R^d.  Finally, they prove no results for SGD, whereas our paper does.
>
>
> (iv) “single learning rate doesn’t apply for all parameters”
> Correct. The algorithm can use a single learning rate for scale-invariant parameters but needs a tuned rate for the scale-variant ones. In feedforward nets, the number of scale-variant parameters scales as the number of nodes and the number of scale-invariant parameters scales as the number of edges (up to weight sharing).  Thus the vast majority of parameters are scale-invariant.
>
>
> (v) “Relation between original loss and loss using BN.”
> Our results hold for the loss of batch-normalized network (“BN-loss”)  which is different from the loss of the original network (“BN-less loss”). Probably the reshaping of loss function due to BN is very important but currently hard to analyse theoretically because we lack a good mathematical understanding of the loss landscape (even BN-less).

---

> > ### Comment · AnonReviewer1 · 2018-11-19
> > **Some Clarification**
> >
> > (i)-(ii)
> > My point was that BN has been never experimentally studied in the asymptotic regime where the results of the paper apply. The shown auto-tuning rate is an interesting property, but there is no evidence that it is relevant to the experimental successes of BN that are mentioned.
> >
> > (iii) Thanks for clarification. The paper of Wu et al. 2018 claims in particular: " The recently proposed batch normalization ... is robust to the choice of Lipschitz constant of the gradient in loss function, allowing one to set a large learning rate without worry". I see now that this work does not make this claim formal and according to the authors' explanation above making it formal takes all the derivations of the submission.
> >
> > (iv) The theoretical advantage of the shown auto-rate tuning is not completely clear. It is not excluded that introducing normalization while easing the learning rate tuning for scale-invariant parameters is making it harder for scale-variant ones. There is a learning rate to tune in the end, no matter how many parameters are scale-invariant.

---

> > > ### Author Response · Authors · 2018-11-27
> > > **Thanks again for your thoughtful review! Experiments are added.**
> > >
> > > Thanks again for your thoughtful review.
> > >
> > > Theory ---almost by definition---may not lead to immediate practical applications. Sometimes the goal is better understanding. That has proved difficult for BN, as described in the introduction.
> > >
> > > Please also note that we are not proposing some new algorithm which could achieve the same test error as existing methods with less tuning, but are trying to understand why BN helps optimization in the training process.
> > >
> > > We've now uploaded a new revision with additional experiments, which exhibits the advantage of auto rate-tuning led by BN in training.

---

### Official Review · AnonReviewer2 · 2018-11-05
**A good paper**

**Rating:** 7
**Confidence:** 2

**Review:**

The paper is well written and easy to follow. The topic is apt.

I don’t have any comments except the following ones.

Lemma 2.4, Point 1: The proof is confusing. Consider the one variable vector case. Assuming that there is only one variable w, then \nabla L(w) is not perpendicular to w in general. The Rayleigh quotient example L(w)  = w’*A*w/ (w’*w) for a symmetric matrix A, then \nabla L(w) = (2/w’*w)(Aw - L(w)*w), which is not perpendicular to w.
Even if we constrain ||w ||_2 = 1, then also  \nabla L(w)  is not perpendicular to w.
Am I missing something?

What is G_t in Theorem 2.5. It should be defined in the theorem itself. There is another symbol G_g which is a constant.

---

> ### Author Response · Authors · 2018-11-18
> **Lemma 2.4 is correct and the issue of G_t is fixed**
>
> Thanks for your positive feedback.
>
> (1). Lemma 2.4, Point 1: The gradient in your example is indeed perpendicular to w which can be seen as follows.
>
> w’ * \nabla L(w) = w’ * (2/w’*w)(Aw - L(w)*w) =  (2/w’*w)(w’Aw - L(w)*(w’*w)) =  (2/w’*w)(w’Aw - w’Aw) = 0.
>
> In case of one variable vector, our proof is to take the derivative of c on both sides of F(w) = F(cw), which is the definition of scale-invariance. Then the left-hand side becomes 0 and the right-hand side becomes w’ * \nabla F(cw)  by chain rule. Taking c = 1, we can conclude that w’ * \nabla F(w)  = 0.
>
> (2). Theorem 2.5: Sorry G_t should be G_t^{(i)}. We will correct this typo in the next revision of this paper.
>
> For t = 0, G_t^{(i)} are all initialized to some value. The recursion formula for G_t^{(i)} is shown in equation (9).

---

> > ### Comment · AnonReviewer2 · 2018-11-23
> > **Got it**
> >
> > Thanks.

---

### Official Review · AnonReviewer4 · 2018-11-13
**Theoretical Analysis of Auto Rate-Tuning by Batch Normalization**

**Rating:** 7
**Confidence:** 2

**Review:**

* Strengths:
- The paper gives theoretical insight into why Batch Normalization is useful in making neural network training more robust and is therefore an important contribution to the literature.
- While the actual arguments are somewhat technical as is expected from such a paper, the motivation and general strategy is very easy to follow and insightful.

* Weaknesses:
- The bounds do not immediately apply in the batch normalization setting as used by neural network practitioners, however there are practical ways to link the two settings as pointed out in section 2.4
- As the authors point out, the idea of using a batch-normalization like strategy to set an adaptive learning rate has already been explored in the WNGrad paper. However it is valuable to have a similar analysis closer to the batch normalization setting used by most practitioners.
- Currently there is no experimental evaluation of the claims, which would be valuable given that the setting doesn't immediately apply in the normal batch normalization setting. I would like to see evidence that the main benefit from batch normalization indeed comes from picking a good adaptive learning rate.

Overall I recommend publishing the paper as it is a well-written and insightful discussion of batch normalization. Be aware that I read the paper and wrote this review on short notice, so I didn't have time to go through all the arguments in detail.

---

> ### Author Response · Authors · 2018-11-27
> **Thanks for your appreciation! Experiments are added.**
>
> Thanks for your valuable review! We've added an experiment section in the new revision, showing how BN helps convergence in the training process.

---

### Author Response · Authors · 2018-11-27
**Experiment results are added in the new revision**

We thank the reviewers for their valuable comments. We uploaded a new revision with additional experiments showing the advantage of auto rate-tuning behavior of BN in training.

Two settings were studied:
1. Training VGG with BN on cifar10, using standard SGD (without momentum, learning rate decay, weight decay).
2.  Training VGG with BN on cifar10, using Projected SGD:  at each iteration, the algorithm first takes a gradient update and then projects each scale-invariant parameter to the sphere with radius equal to its 2-norm before this iteration, i.e., rescales each scale-invariant parameter such that they maintain their norms during training.

In both settings, learning rate for scale variant parameter is 0.1 but rates for scale invariant parameters vary from 0.01 to 100, a very large range. The plots show that in setting 1 the training loss of SGD lways  gets very small, while in setting 2, the training loss of PSGD remains large for lr > 1.

The only difference between SGD and PSGD is that the implicit rate-tuning behavior on scale invariant parameters is blocked because of the fixed norm of scale-invariant parameters. So we can conclude that the auto rate-tuning phenomenon does happen here and it helps convergence in training when learning rate is large.

---

### Meta-Review · Area_Chair1 · 2018-12-12
**Good theoretical contribution to understanding batch normalization.**

**Confidence:** 4
**Recommendation:** Accept (Poster)

**Metareview:**

This paper conducted theoretical analysis of the effect of batch normalisation to auto rate-tuning. It provides an explanation for the empirical success of BN. The assumptions for the analysis is also closer to the common practice of batch normalization compared to a related work of Wu et al. 2018.

One of the concerns raised by the reviewer is that the analysis does not immediately apply to practical uses of BN, but the authors already discussed how to fill the gap with a slight change of the activation function. Another concern is about the lack of empirical evaluation of the theory, and the authors provide additional experiments in the revision. R1 also points out a few weaknesses in the theoretical analysis, which I think would help improve the paper further if the authors could clarify and provide discussion in their revision.

Overall, it is a good paper that will help improve our theoretical understanding about the power tool of batch normalization.